# Lipofuscin-like autofluorescence within microglia and its impact on studying microglial engulfment

Jacob M. Stillman[1,2,4], Francisco Mendes Lopes[1,4], Jing-Ping Lin ⓘ [3], Kevin Hu[3], Daniel S. Reich ⓘ [3] & Dorothy P. Schafer ⓘ [1] ✉

Engulfment of cellular material and proteins is a key function for microglia, a resident macrophage of the central nervous system (CNS). Among the techniques used to measure microglial engulfment, confocal light microscopy has been used the most extensively. Here, we show that autofluorescence (AF) likely due to lipofuscin (lipo-AF) and typically associated with aging, can also be detected within microglial lysosomes in the young mouse brain by light microscopy. This lipo-AF signal accumulates first within microglia and it occurs earliest in white versus gray matter. Importantly, in gray matter, lipo-AF signal can confound the interpretation of antibody-labeled synaptic material within microglia in young adult mice. We further show that there is an age-dependent accumulation of lipo-AF inside and outside of microglia, which is not affected by amyloid plaques. We finally implement a robust and cost-effective strategy to quench AF in mouse, marmoset, and human brain tissue.

Microglia are highly phagocytic tissue-resident macrophages of the central nervous system (CNS). While the phagocytic activity of microglia has historically been attributed to clearing dead or dying cells, the list of microglial phagocytic substrates has expanded in recent years to include synaptic material[1–4], extracellular matrix proteins[5], and protein aggregates (amyloid beta, tau, etc.)[6]. From this work, the engulfment of cellular and protein material by microglia has been shown to regulate synaptic connectivity and modulate neurological disease phenotypes[1–4]. Microglial engulfment is also an emerging target for therapeutic intervention in diseases ranging from Alzheimer's disease to schizophrenia[7–9]. Therefore, it is critical that the analysis of microglial engulfment of cellular and protein substrates is performed with the highest rigor.

Confocal light microscopy has become a standard method to measure microglial engulfment function in tissues and cells[10,11]. A potential confound of these studies is autofluorescence (AF) in brain tissue, particularly in aging. Likely the largest source of AF in tissues is lipofuscin. Lipofuscin is a mixture of highly oxidized lipids, misfolded proteins, and metals, which accumulates with age within lysosomal compartments[12–14]. These lipofuscin aggregates autofluoresce across the fluorescent spectrum, making it challenging to image fluorescently labeled cells and molecules by light microscopy[15–18]. In microglia, the aggregation of lipofuscin can be induced by incomplete myelin digestion and disruption of the lysosomal pathway, which implicates phagocytosis of cellular material as a key mechanism leading to increased lipofuscin burden[19]. Further, lipofuscin accumulation in microglia is an age-dependent process and it has been estimated that AF-positive microglia, which is likely lipofuscin, outnumber AF-negative microglia by greater than two-fold in 6-month-old mice[17,20]. However, recently AF attributed to lipofuscin has been shown within microglia lysosomes in mice as early as 7–9 weeks[17,18]. Thus, it is important to consider the potential confound that lipofuscin in microglia can be misinterpreted as engulfed cellular and protein material.

Here, we assessed AF, which is likely from lipofuscin (we term lipo-AF), within microglia using confocal light microscopy across the

[1]Department of Neurobiology, Brudnick Neuropsychiatric Research Institute, University of Massachusetts Chan Medical School, Worcester, MA 01605, USA. [2]University of Massachusetts Chan Morningside Graduate School of Biomedical Sciences, Neuroscience Program, Worcester, MA, USA. [3]Translational Neuroradiology Section, National Institute of Neurological Disorders and Stroke, National Institutes of Health, Bethesda, MD 20892, USA. [4]These authors contributed equally: Jacob M. Stillman, Francisco Mendes Lopes. ✉e-mail: dorothy.schafer@umassmed.edu

developing, adult, aged and diseased mouse brain. Our data show that microglia are the first resident CNS cell type to accumulate lipo-AF in mouse brain, and this accumulation occurs earliest in white matter. We show that lipo-AF inside and outside of microglia accumulates with aging; however, microglial lipo-AF is not increased in plaque associated microglia during neurodegeneration in mouse brain. We also provide evidence that, if not taken into consideration, lipo-AF can be misinterpreted as engulfed material within microglia, even in the young, adult brain. Finally, we provide an adaptable, cost-effective pre-staining AF quenching protocol that preserves immunofluorescent antibody signal. This protocol can further be applied across species, including mouse, marmoset, and human brain tissue.

## Results

### Lipofuscin-like autofluorescence accumulates first within microglia

We began imaging tissue at postnatal day 90 (P90) when other studies have shown a significant accumulation of AF, likely due to lipofuscin, within microglia in the mouse brain by light microscopy[17,18,21]. We focused our initial analyses in the somatosensory cortex and neighboring visual and auditory cortices and imaged unstained tissue. The AF signal within the unstained cortex at P90 was observed with a 488 nm laser (Green, Band Pass Filter (BP) 525/50), 561 nm laser (Red; BP 629/62), and 638 nm laser (Far-red; BP 690/50), but not the 405 nm laser (Blue, BP 450/50) (Fig. 1a–c). Using the 561 nm laser line, we further show that AF could be observed at multiple relative laser intensities and that its signal intensity increased accordingly (Fig. 1d–f). Next, we imaged cortical tissue co-stained with a microglia marker (anti-P2RY12) and lysosomal marker (anti-CD68) (Fig. 1g) and found that at P90, 26.3% of total AF volume was microglial (Fig. 1g, h) and, within microglia, 80.0% was localized to lysosomal (CD68+) compartments (Fig. 1g, i). Using different confocal microscopes, we

found a similar ratio of microglial/non-microglial AF in the P90 mouse cortex (Supplementary Fig. 1). Because of the observed excitation and emission properties of the AF signal as well as its localization to lysosomes, we reasoned this AF is most likely due to lipofuscin[13,16,18,21–23]. Other AF molecules generally have a tighter excitation and emission spectra and are not localized specifically to lysosomal compartments[23]. However, because there does not exist a highly specific stain for lipofuscin, we refer to it as lipofuscin-like AF or lipo-AF.

We then extended our analyses to earlier developmental timepoints and compared lipo-AF within and outside microglia (anti-IBA1) and within their lysosomes (anti-CD68), in the somatosensory cortex (Fig. 2 and Supplementary Fig. 2). We first observed CD68 within all microglia in the somatosensory cortex, but it was highest at early developmental time points (P5) and with aging (24 m) (Supplementary Fig. 2c). Interestingly, despite a higher lysosomal load within microglia in the young postnatal cortex, there was small to negligible amounts (0–0.1% of microglia volume) of lipo-AF within these lysosomal compartments (Fig. 2a, b and Supplementary Fig. 2b). However, by P60-P90, nearly 100% of microglia accumulated a small (0.1–1% of microglia volume) to moderate amount (1–2% of microglia volume) of lipo-AF (Fig. 2a, b and Supplementary Fig. 2b). As expected, nearly 100% of microglia from the aged, 24-month-old brain had high amounts (>2.0% of microglia volume) of lipo-AF burden (Fig. 2a, b and Supplementary Fig. 2b). Interestingly, most observed lipo-AF in the cortex was localized within microglia at earlier ages (≤P60), but a significantly higher percentage of lipo-AF was observed outside of microglia at ≥P90 (Fig. 2c). Of note, all experiments described above were performed with a 561 nm laser (Red; BP 629/62). A similar time course of lipo-AF accumulation within and outside gray matter microglia was observed with the 488 nm laser (BP 525/50) and 638 nm laser (BP 690/50) lines (Supplementary Fig. 3). Additionally, the majority of non-microglial AF that began to accumulate in the older cortex was largely localized to

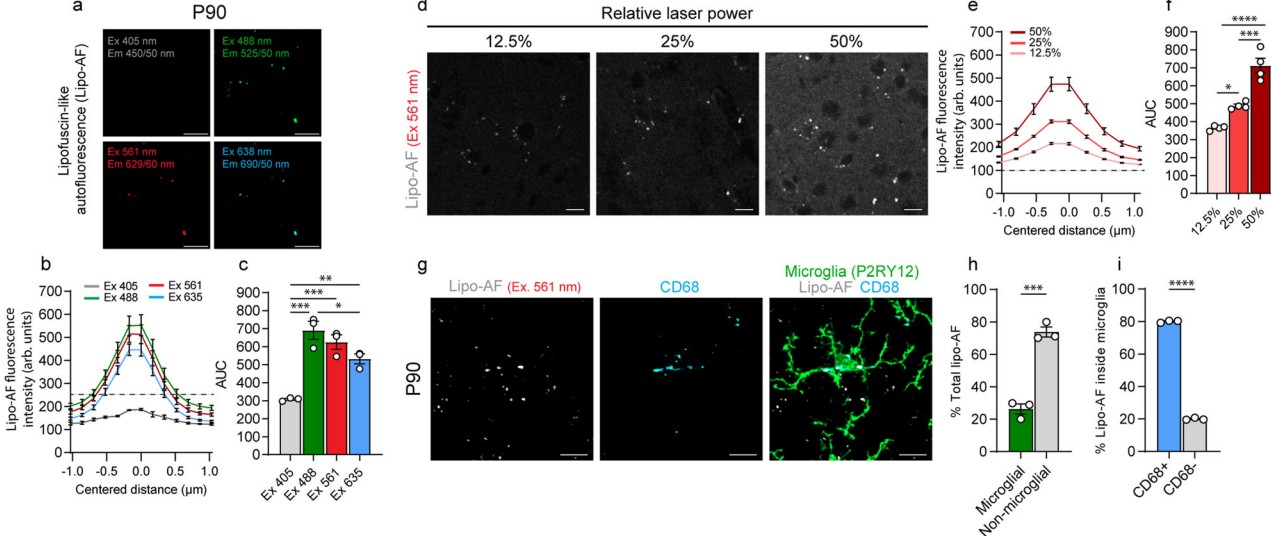

**Fig. 1 | Lipofuscin-like autofluorescence (lipo-AF) in the P90 mouse cortex.**
**a** Representative images of lipo-AF signal in the young adult P90 somatosensory cortex. Excitation (Ex) and Emission (Em) for each laser line are indicated. Scale bar = 5 μm. **b** Quantification of lipo-AF fluorescence intensity measured in arbitrary (arb.) units of a lipo-AF puncta for each laser line. **c** Quantification of area under the curve from (**b**). $n = 3$ (2 F, 1 M) mice. One-way ANOVA with Tukey's multiple comparisons test ($F = 22.97$, df = 11; Ex. 405 vs 488 $p = 0.0003$, 405 vs 561 $p = 0.0009$, 405 vs 635 $p = 0.0081$, 488 vs 635 $p = 0.0495$). **d** Representative images of lipo-AF across different 561 nm laser intensities in the young adult somatosensory cortex. Scale bar = 10 μm. **e** Quantification of lipo-AF fluorescence intensity across of a lipo-AF puncta for each laser line. $n = 4$ mice (3 M, 1 F). **f** Quantification of area under the curve (AUC) in (**e**). $n = 4$ (3 M, 1 F) mice. One-way ANOVA with Tukey

multiple comparisons test ($F = 50.46$, df = 11; 12.5% vs 25% $p = 0.0163$, 12.5% vs 50% $p = <0.0001$, 25% vs 50% $p = 0.0003$). **g** Representative image of an anti-P2RY12 immunolabelled microglia in the P90 mouse somatosensory cortex containing lipo-AF within anti-CD68+ lysosomal compartments. Scale bars = 5 μm.
**h** Quantification of the percentage of lipo-AF within and outside microglia. Data are represented as mean ± SEM, $n = 3$ (2 F, 1 M) mice. Two-tailed unpaired $t$ test ($t = 84.75$, df = 4; $p = 0.0004$). **i** Quantification of the percentage of lysosomal (CD68+) and non-lysosomal (CD68-) lipo-AF inside microglia. $n = 3$ (2 F, 1 M) mice. Two-tailed unpaired $t$ test ($t = 10.75$, df = 4; $p = < 0.0001$). *$p < 0.05$ **$p < 0.01$ ***$p < 0.001$ ****$p < 0.0001$. All data presented as mean ± SEM. Source data are provided as a Source Data file.

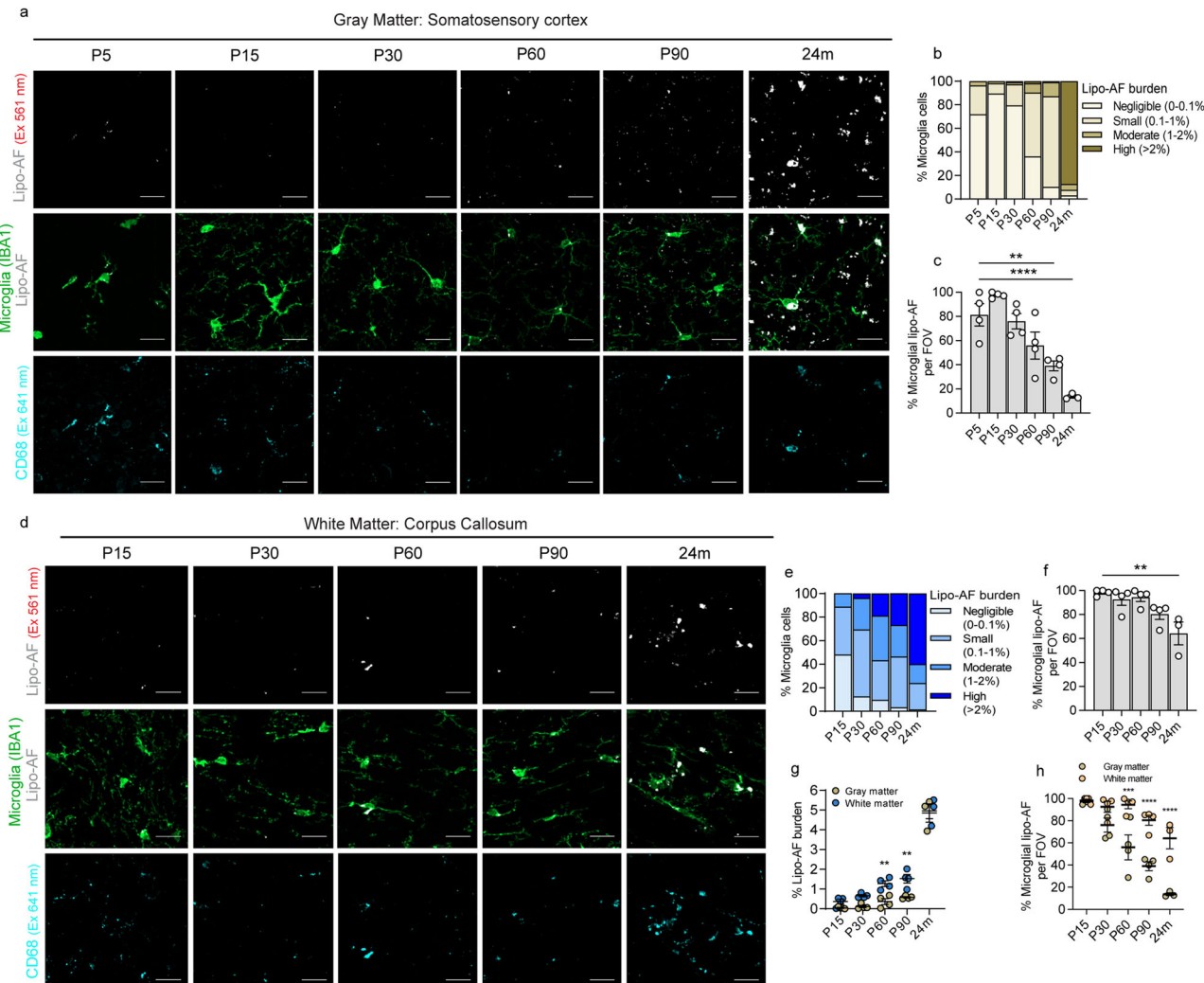

**Fig. 2 | Lipo-AF accumulation in microglia in the gray and white matter.**
Representative images of anti-IBA1+ microglia, lipo-AF (excitation (Ex) 561 nm) and anti-CD68+ lysosomes in the developing (P5, P15, P30), young adult (P60, P90), and aged (24 months) gray matter (**a**, cortex) or white matter (**d**, corpus callosum). Scale bars = 10 μm. Quantification of the percentage of cortical microglial with negligible (0–0.1%), small (0.1–1%), moderate (1–2%) and high (>2%) percentages of their total volume occupied by lipo-AF in the gray (**b**) or white (**e**) matter. Quantification of the percentage of lipo-AF per field of view (FOV) inside microglia in the gray (**c**) and white (**f**) matter. **c** One-way ANOVA with Dunnett's multiple comparisons test ($F = 17.46$, df = 22; P5 vs P90 $p = 0.0020$, P5 vs 24 m $p < 0.0001$). **f** One-way ANOVA with Dunnett's multiple comparisons test ($F = 7.225$, df = 18; P15 vs

24 m $p = 0.0012$). **g** Quantification of the average percentage of microglial volume occupied by lipo-AF in gray and white matter. Two-way ANOVA with Šidák's multiple comparisons test (Interaction: $F = 1.877$, df = 4; Row Factor: $F = 190.5$, df = 4; Column Factor: $F = 23.81$, df = 1; P60 $p = 0.0058$, P90 $p = 0.0040$). **h** Quantification of the average percentage of lipo-AF per field of view inside of microglia in gray and white matter microglia. Two-way ANOVA with Šidák's multiple comparisons test (Interaction: $F = 6.267$, df = 4; Row Factor: $F = 29.18$, df = 4; Column Factor: $F = 66.24$, df = 1; P60 $p = 0.0002$, P90 & 24 m $p < 0.0001$); *$p < 0.05$ **$p < 0.01$ ***$p < 0.001$ ****$p < 0.0001$. All data are represented as mean ± SEM, $n = 3$–4 mice per experiment, 2 M, 2 F for each timepoint between P5-P90. 2 M, 1 F for 24 m. Source data are provided as a Source Data file.

neurons and a small portion was localized to anti-LYVE1-positive brain border macrophages (Supplementary Fig. 2f–i).

To assess regional differences, we compared the gray matter-enriched somatosensory cortex to the white matter-enriched corpus collosum (Supplementary Fig. 2a). For white matter analyses, we began analyzing at P15 when myelination is underway. First, we observed an age-dependent increase in CD68+ lysosomes in white matter microglia (Supplementary Fig. 2e). However, unlike gray matter, there appeared to be earlier accumulation of lipo-AF within microglia in the white matter with nearly all microglia accumulating small to moderate amounts (0.1–1% and 1–2% of microglia volume, respectively) of lipo-AF by P30 (Fig. 2d–f and Supplementary Fig. 2d), and a significant increase in lipo-AF within microglia in white matter at P60 compared to gray matter (Fig. 2a, b, g and Supplementary Fig. 2b). Also, in

contrast to gray matter, most lipo-AF was restricted to microglia in white matter at all ages analyzed (Fig. 2f, h).

Together, these data demonstrate that microglia are the first cells to accumulate lipo-AF in gray and white matter, this lipo-AF accumulation occurs earlier than previously appreciated, and accumulation of lipo-AF within microglia occurs earliest in the white matter. In addition, most lipo-AF is largely restricted to microglia throughout the lifespan in white matter, while lipo-AF begins to accumulate more in neurons with aging in the gray matter. These data suggest that there is a specific environment in white matter that is conducive to early accumulation of lipo-AF specifically in microglia, which is supported by published work that microglial engulfment of myelin is highly correlated with the accumulation of lipo-AF in microglia[12,19].

## Lipo-AF within gray matter microglia is unchanged in the presence of amyloid plaques

Besides engulfment of myelin, increased microglial engulfment of cellular material and protein aggregates has also been shown in the context of neurodegeneration. Additionally, aging is one of the biggest risk factors for developing a neurodegenerative disease and lipofuscin is a well-established cellular phenotype in aging[22]. We, therefore, hypothesized that microglial lipo-AF accumulation would be exacerbated by neurodegeneration. We tested this hypothesis using the Alzheimer's disease (AD)-relevant mouse model of neurodegeneration, the 5xFAD model (Fig. 3). Surprisingly, while AF was observed in microglia in the somatosensory cortex of 9-month-old 5xFAD mice, it was comparable to 9-month wild-type (WT) controls (Fig. 3a, b). There was also no statistical difference in the amount of lipo-AF in ThioS+ plaque-associated versus plaque-unassociated microglia in the 5xFAD cortex (Fig. 3b) and there was no significant correlation between plaque size and lipo-AF burden within microglia (Fig. 3c). Thus, these data demonstrate that microglia accumulate lipo-AF similarly in 9 month 5xFAD mice as 9 month wild-type controls and the degree of lipo-AF within microglia is not impacted by amyloid plaque size.

## Lipo-AF co-localizes with synaptic protein inside microglia in the adult gray matter

Given that lipo-AF is localized within microglial lysosomes (Fig. 1g, i), it could confound assessments of engulfed cellular material within microglia. As synapses are key phagocytic substrates for microglia in

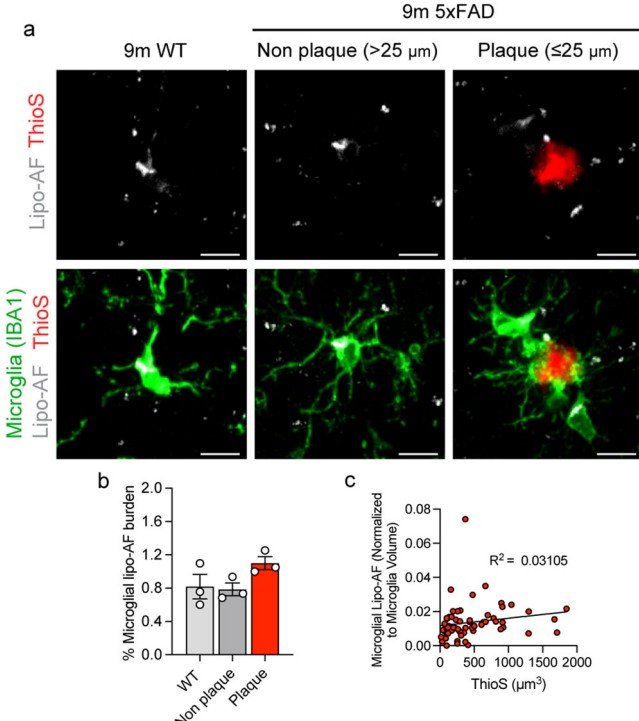

**Fig. 3 | Lipo-AF in microglia during Alzheimer's disease-related neurodegeneration.** **a** Representative images of anti-IBA1 immunolabelled microglia, Thioflavin-S labeled amyloid β plaques (ThioS), and lipo-AF in the somatosensory cortex of 9-month-old 5xFAD mice and wild-type (WT) littermates. Scale bars = 5 μm. **b** Quantification of total microglial volume occupied by lipo-AF in plaque-associated microglia, non-plaque associated microglia and microglia from WT littermates. Data are represented as mean ± SEM, $n = 3$ mice. One-way ANOVA with Tukey multiple comparisons test ($F = 10.82$, df = 8). **c** ThioS+ plaque volume correlated with total microglial lipo-AF volume normalized to microglial volume surrounding the plaque in the 5XFAD model. $n = 61$ individual plaques. Linear regression ($F = 1.81$; DFn = 1, DFd = 59; $R^2 = 0.03105$). (1 M, 2 F 5xFAD. 3 M, WT). Source data are provided as a Source Data file.

health and disease[1–4], we next determined the potential of lipo-AF to confound microglial synapse engulfment analysis. Sections of P5-P90 mouse somatosensory cortex were immunostained for anti-IBA1 to label microglia and anti-vesicular glutamate transporter 2 (VGluT2) to label excitatory presynaptic terminals in cortical layer IV (Fig. 4a). We then imaged anti-IBA1 with the 488 nm laser line, anti-VGluT2 with the 561 nm laser line, and lipo-AF with the 635 nm laser line. We found a significant increase in colocalization of anti-VGluT2 signal with lipo-AF inside microglia by P60, which extended to P90 (Fig. 4a, b). There was significantly less colocalization of the anti-VGluT2 signal with lipo-AF outside of microglia (Fig. 4c). We further compared the fluorescence intensity of anti-VGluT2 to lipo-AF signal. P90 sections of mouse somatosensory cortex were again immunostained for anti-VGluT2. Neighboring sections from the same brain were left unstained to measure lipo-AF in the same region and with the same laser line (561 nm laser, BP 629/62). The intensity of the anti-VGluT2 immuno-labelled puncta outside and inside the microglia boundaries, as well as lipo-AF in the neighboring sections, were measured in resulting images (Fig. 4d, e). Notably, at P90, the anti-VGluT2 puncta intensity inside the microglia was a comparable intensity to the lipo-AF intensity (Fig. 4e). We obtained similar results using another presynaptic marker anti-VGluT1 (Fig. 4f, g). Interestingly, when we repeated these experiments with the postsynaptic marker anti-Homer1, there was a significant difference in the fluorescence intensity of internalized anti-Homer 1 compared to lipo-AF (Fig. 4h, i). These data raise the possibility that lipo-AF can be confused for engulfed synaptic material, depending on the antibody or cellular material that is analyzed.

## Photobleaching can be used to eliminate lipo-AF signal prior to microglial engulfment analyses

Considering the potential for lipo-AF to confound downstream analyses of engulfed material within microglia, we explored protocols to reduce lipo-AF signal in brain tissue. Previous groups have used a commercially available derivative of Sudan Black to eliminate AF in tissues[18]. We used a next generation version of this reagent called TrueBlack Plus™ and imaged the mouse somatosensory cortex at P90, a timepoint when lipo-AF accumulation in microglia is significantly increased (Fig. 2). Using the 561 nm laser (BP 629/62), we identified a significant decrease in lipo-AF intensity within the field of view and a significant reduction in lipo-AF accumulation within microglia with the TrueBlack Plus™ protocol (Fig. 5a–d). However, this protocol also resulted in a significant decrease in the intensity of immunostained P2RY12 (Fig. 5e). We, therefore, took steps to improve this methodology using a commercially available MERSCOPE photobleacher device typically used for multiplexed error-robust fluorescence in situ hybridization (MERFISH), which uses LED-based lights to photobleach samples. After incubating mouse brain sections in photobleaching light for 12 h, we proceeded with our standard immunostaining protocol (Fig. 5f). Similar to chemical quenching, this photobleaching method significantly eliminated lipo-AF signal within P90 microglia (Fig. 5f–i), but without compromising the fluorescent signal of anti-P2RY12 (Fig. 5j).

When next used this photobleaching protocol to more directly test the extent to which the anti-VGluT2 signal detected within the microglia boundaries could be confounded by lipo-AF. We photobleached P5 and P90 tissue to eliminate lipo-AF signal and then immunostained tissue for anti-VGluT2 and anti-IBA1. We then imaged lipo-AF within the somatosensory cortices with a 561 nm laser (BP 629/62). We chose to compare P90 to P5 as lipo-AF is significantly increased at P90 and P5 was a developmental timepoint where lipo-AF was low (Figs. 1 and 2). Also, P5 is an age where the somatosensory cortex is known to undergo extensive experience-dependent synapse remodeling by phagocytic microglia[24]. In untreated sections, apparent engulfed VGluT2 material was detectable within microglia at P5 and P90 (Fig. 5k–n). However, after photobleaching, this engulfed VGluT2

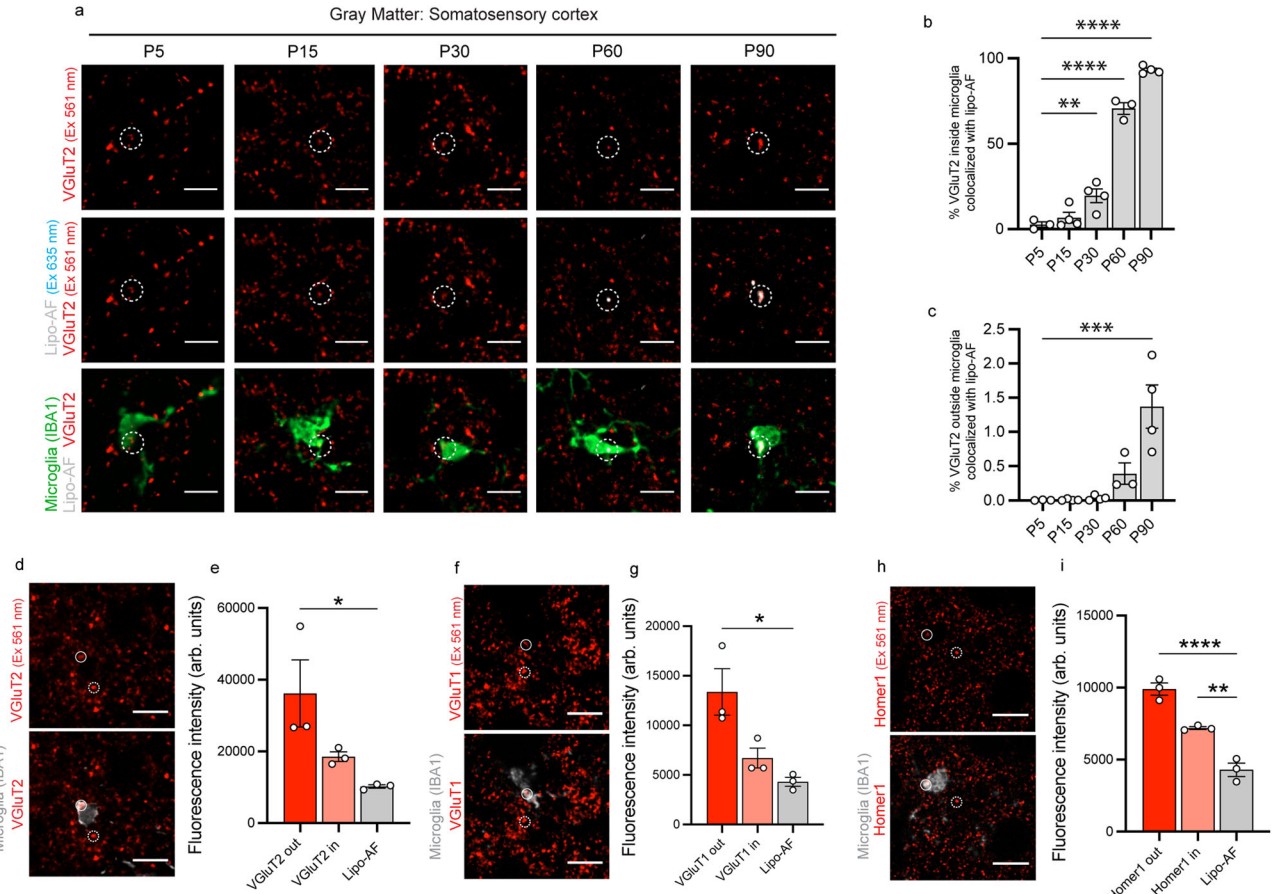

**Fig. 4 | Microglial Lipo-AF co-localizes with engulfed synaptic material within microglia in the young adult mouse brain. a** Representative images of anti-VGluT2+ synaptic material excited (Ex) by a 561 nm laser, lipo-AF excited by a 635 nm laser and anti-IBA1+ microglia excited by a 488 nm laser in layer IV of the somatosensory cortex. Circles highlight engulfed anti-VGluT2+ synaptic material. Scale bar = 10 μm. **b** Quantification of the volume of VGluT2+ material colocalized with lipo-AF inside microglia., $n$ = 3–4 mice (2 M, 2 F for each for P15, P30, P90; 2 M, 1 F for P5 and P60). One-way ANOVA with Dunnett's multiple comparisons test ($F$ = 192.1, df = 17, P5 vs P30 $p$ = 0.0060, P5 vs P60 & P90 $p$ < 0.0001). **c** Quantification of the volume of VGluT2+ material outside anti-IBA1+ microglia co-localized with lipo-AF. $n$ = 3–4 mice (2 M, 2 F for each for P15, P30, P90; 2 M, 1 F for P5 and P60). One-way ANOVA with Dunnett's multiple comparisons test ($F$ = 13.01, df = 17, $p$ = 0.0003). Representative image of anti-IBA1+ microglia with anti-VGluT2+ (**d**), anti-VGluT1+ (**f**), and anti-Homer1+ (**h**) synapses in the P90 somatosensory cortex. Scale bar = 10 μm. Quantification of the fluorescence intensity in arbitrary (arb.) units of total apparent VGluT2+ (**e**), VGluT1+ (**g**), or Homer1+ (**i**) puncta outside of anti-IBA1+ microglia (VGluT2 out, VGluT1 out, or Homer1 out) and inside of anti-IBA1+ microglia (VGluT2 in, VGluT1 in, Homer1 in) in the 561 nm channel and lipo-AF signal intensity in the 635 nm channel. **e** $n$ = 3 mice (2 M, 1 F). One-way ANOVA with Tukey's multiple comparisons test ($F$ = 5.873, df = 8; VG2 out vs Lipo-AF $p$ = 0.0353). **g** $n$ = 3 mice (3 M). One-way ANOVA with Tukey's multiple comparisons test ($F$ = 9.090, df = 8; VG1 out vs Lipo-AF $p$ = 0.0121). **i** $n$ = 3 mice (3 M). One-way ANOVA with Tukey's multiple comparisons test ($F$ = 58.93, df = 8); *$p$ < 0.05 **$p$ < 0.01 ***$p$ < 0.001 ****$p$ < 0.0001. All data are represented as mean ± SEM. Source data are provided as a Source Data file.

signal was no longer detected at P90 (Fig. 5m, n). In contrast, microglia within the P5 cortex displayed similar levels of engulfed VGluT2+ material in the photobleached and non-photobleached condition (Fig. 5k, l). Taken together, these data suggest that lipo-AF can confound the interpretation of fluorescent signal within microglia in young adult mouse cortex, but this is less of a concern in neonate tissue. Therefore, particularly in adult brain, precautions should be taken to eliminate lipo-AF to avoid false positive detection of engulfment events.

## Commercial and homemade photobleaching devices effectively eliminate autofluorescence across species

In addition to the commercial MERSCOPE photobleaching device that uses LED-based lighting for photobleaching tissues, we designed a more cost-effective option based on materials that can be widely obtained (Supplementary Fig. 4). Using this homemade device or the commercial device, 24 h of photobleaching effectively quenched lipo-AF in all laser lines in 24 month-old mouse cortex (Fig. 6a–c). We next extended this protocol to other species and fixation and storage

conditions. Similar to the aged mouse brain, photobleaching for 24 h significantly reduced the signal intensity from AF across multiple fluorescence channels in formalin-fixed, paraffin-embedded (FFPE) 11–13 year-old marmoset cortex (Fig. 6d–f) and FFPE 60–77 year-old human cortex from multiple sclerosis (MS) subjects (Fig. 6g–i). We also emphasize that each of these tissues were fixed for shorter (hours in mouse) or longer times (days in marmoset and human). They have also been stored for different times, including days/weeks (mouse) to years (marmoset and human), and they have been prepared either by freezing methods (mouse) or by paraffin embedding (marmoset and human). We further immunostained tissues following photobleaching and observed no significant effect on the immunofluorescent signal intensity in mouse, marmoset, or human brain tissues (Supplementary Fig. 5). Thus, these results demonstrate a cost-effective pre-staining protocol that reliably eliminates AF in tissue sections across different tissue preparations and storage conditions without compromising immunofluorescent signal. As this can be adapted for multiple species, including human, the protocol has broad applicability.

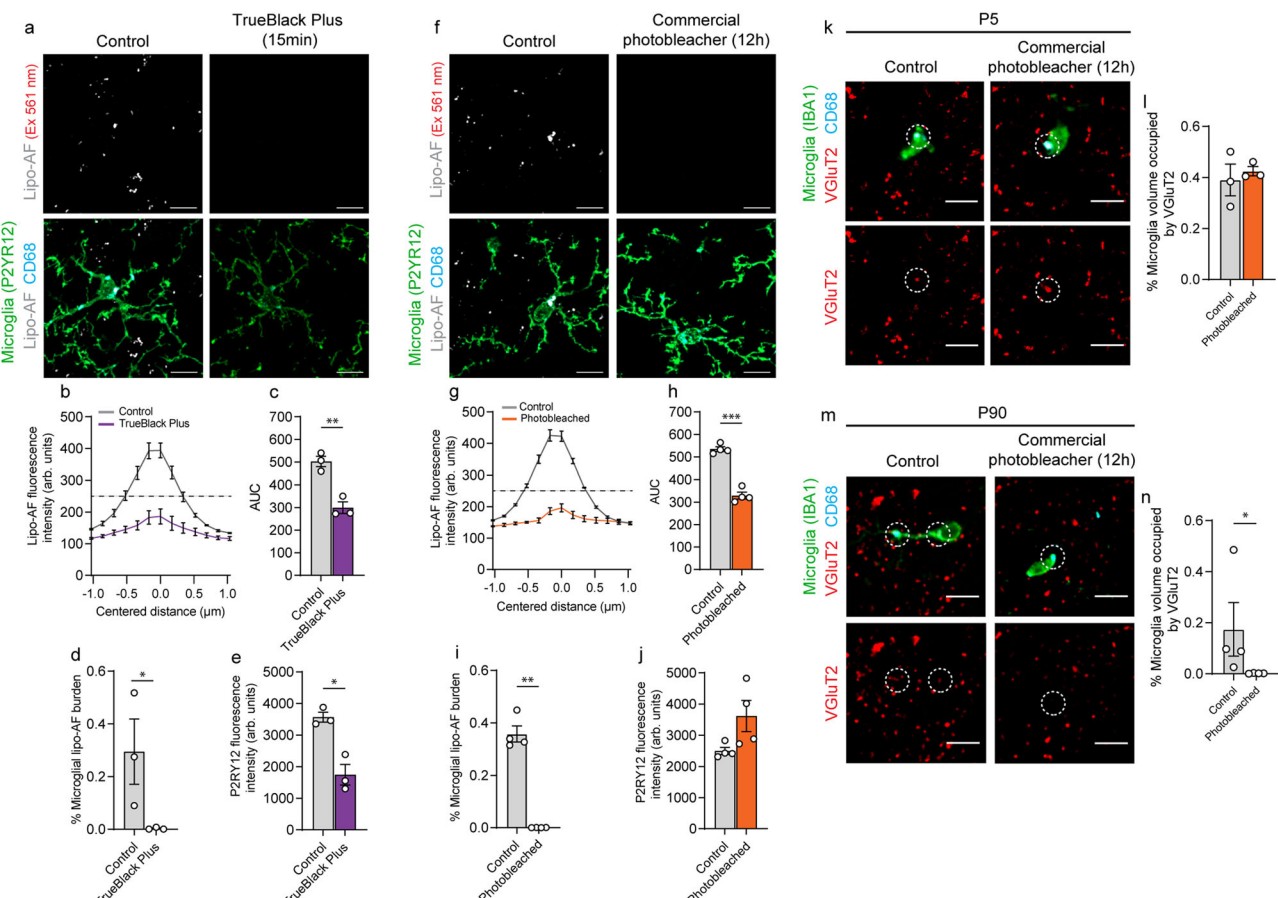

**Fig. 5 | Lipo-AF can be mistaken for engulfed synaptic material within microglia.** Representative images of anti-P2RY12+ microglia and lipo-AF in TrueBlack Plus (TBP)-treated (**a**) or photobleached (**f**) samples from P90 mouse somatosensory cortex. Laser excitation (Ex) is indicated for the lipo-AF. Scale bars = 5 μm. Quantification of lipo-AF intensity for each laser line following TBP (**b**) or photobleaching (**g**). Data presented as mean ± SEM, $n$ = 3 mice. **c**, **h** Quantification of area under the curve (AUC) in (**b** and **g**). **c** $n$ = 3 mice. Paired two-tailed $T$ test ($t$ = 14.32, df = 2, $p$ = 0.0048). **h** $n$ = 3 mice. Paired two-tailed $T$ test ($t$ = 22.7, df = 3, $p$ = 0.0002). Quantification of the percentage of microglial volume occupied by lipo-AF following TBP (**d**) or photobleaching (**i**). **d** $n$ = 3 mice. Ratio-paired two-tailed $T$ test ($t$ = 5.998, df = 2, $p$ = 0.0267). **i** $n$ = 3 mice. Paired two-tailed $T$ test ($t$ = 11.53, df = 3, $p$ = 0.0014). Quantification of anti-P2RY12 fluorescence intensity in arbitrary (arb.)

units following TBP (**e**) or photobleaching (**j**). **e** $n$ = 3 mice. Paired two-tailed $T$ test ($t$ = 4.262, df = 2, $p$ = 0.0509). **j** $n$ = 3 mice. Paired two-tailed $T$ test ($t$ = 4.562, df = 3). Representative images of anti-VGluT2+ presynaptic terminals within layer IV of the P5 (**k**) or P90 (**m**) somatosensory cortex with (right) and without (left) photobleaching. Dashed circles indicate anti-VGluT2 signal inside anti-CD68+ microglial lysosomes. Bottom images are the same image without anti-IBA1 immunostaining. Scale bars = 5 μm. Quantification of VGluT2+ material volume within microglial with and without photobleaching at P5 (**l**) and P90 (**n**). **l** $n$ = 3 mice. Paired two-tailed $T$ test ($t$ = 0.7439, df = 2). **n** $n$ = 4 mice. Mann–Whitney two-tailed test. Data presented as ±SEM. *$p \leq 0.05$ **$p < 0.01$ ***$p < 0.001$. (3 M for all TBP experiments; 2 M, 2 F for all photobleaching experiments; 2 M, 2 F for P90; 2 M, 1 F for P5). Source data are provided as a Source Data file.

## Discussion

Here, we assessed microglia-associated lipo-AF in the developing, adult, aged, and diseased mouse brain using confocal light microscopy. We show that microglia are the first cells to accumulate lipo-AF in the white and gray matter and white matter microglia were the first to accumulate lipo-AF. Lipo-AF within and outside microglia increased with aging in gray matter, but was not increased with Alzheimer's disease-related neurodegeneration. Notably, in the gray matter, lipo-AF within microglia can be mis-interpreted as engulfed synaptic material in the young, adult mouse brain. Finally, we provide a cost-effective protocol to rid tissues of AF signal before immunostaining, which reduces the confound of lipo-AF for microglial engulfment studies. Importantly, this protocol can also be applied to any other study reliant on fluorescence light microscopy across species, including in human tissue.

What is the source of lipo-AF within microglia? Similar to retinal pigment epithelium[15], we speculate that lipo-AF is lipofuscin and it is generated due to microglia-mediated phagocytosis of cellular material throughout the lifespan. Consistent with this idea, it has been suggested that engulfment of myelin during demyelination drives lipofuscin

accumulation in microglia[19]. Interestingly, we observed lipo-AF accumulate first and most robustly in white vs. gray matter microglia. We suspect this is due to the continuous remodeling of the myelin sheath throughout the lifespan by microglia, which engulf excess or shed myelin[25–28]. Also, the lipid-rich properties of the myelin membrane may be more conducive to lipofuscin accumulation, which is enriched in oxidized lipids[12]. An important future direction will be to understand the prevalence of this biology in microglia across different diseases and to determine how different phagocytic substrates in gray and white matter may induce the accumulation of lipo-AF differentially in microglia.

In the current study, we also show that microglia accumulate lipo-AF before other CNS cell types. The broad phagocytic role of microglia in early development[1] may explain the presence of the small amount of lipo-AF within microglia in the developing brain. However, this lipo-AF within microglia increases substantially in the adult and aged brain and it co-localizes with synaptic immunostaining within microglia in the adult brain. It is possible that the lysosomal degradation capacity of microglia in the developing brain may be able to maintain lysosomal homeostasis without accumulating a larger lipofuscin load, but this lysosomal capacity changes with aging[12]. In support of this idea,

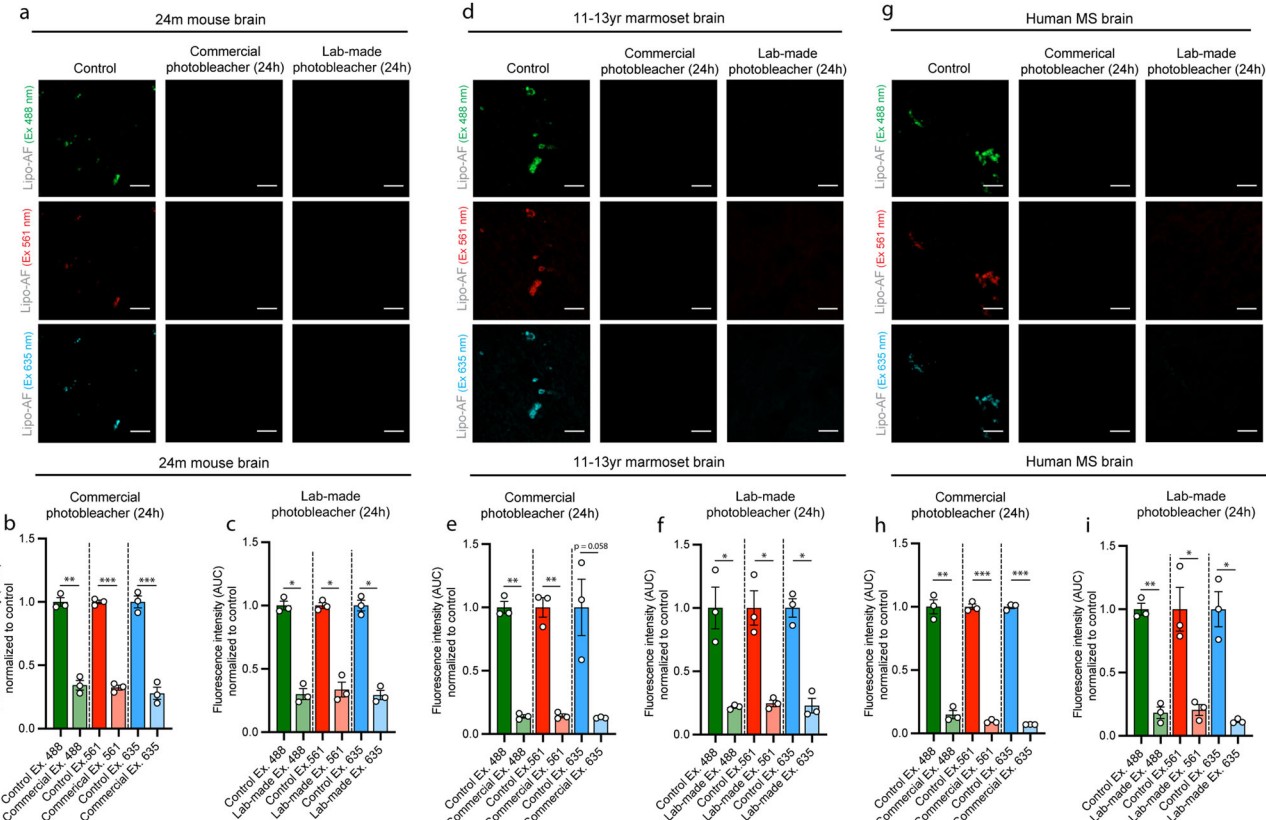

**Fig. 6 | A lab-made photobleacher to quench lipo-AF signal.** Representative images of lipo-AF excited (Ex.) with differ laser lines in untreated or photobleached samples in the aged mouse (**a**), marmoset (**d**), and human multiple sclerosis (MS) (**g**) cortices. Scale bars = 10 μm. **b**–**i** Quantification of lipo-AF puncta fluorescence intensity area under the curve (AUC) in aged mouse (**b**, **c**), marmoset (**e**, **f**), and human (**h**, **i**) cortices without (control, dark bars) or with photobleaching (pale bars) for the are data using the commercial (**b**, **e**, **h**) and lab-made (**c**, **f**, **i**) photobleachers. All data presented as mean ± SEM normalized to control, *n* = 3 mice, 3 marmoset, or 3 human samples. Paired two-tailed *T* tests between laser lines. Ex. 488, *t* = 11.96, df = 2, *p* = 0.0069; (**b**) Ex. 561, *t* = 49.61, df = 2, *p* = 0.0004; (**b**) Ex. 635, *t* = 75.28, df = 2, *p* = 0.0002; (**c**) Ex. 488, *t* = 9.497, df = 2, *p* = 0.0109; (**c**) Ex. 561, *t* = 9.372, df = 2, *p* = 0.0112; (**c**) Ex. 635, *t* = 9.053, df = 2, *p* = 0.0120. **e** Ex. 488, *t* = 24.12, df = 2, *p* = 0.0017; (**e**) Ex. 561, *t* = 12.81, df = 2, *p* = 0.0060; (**e**) Ex. 635, *t* = 3.964, df = 2, *p* = 0.0581; (**f**) Ex. 488, *t* = 4.468, df = 2, *p* = 0.0466; (**f**) Ex. 561, *t* = 5.652, df = 2, *p* = 0.0299; (**f**) Ex. 635, *t* = 5.927, df = 2, *p* = 0.0273. **h** Ex. 488, *t* = 28.95, df = 2, *p* = 0.00012; (**h**) Ex. 561, *t* = 69.05, df = 2, *p* = 0.0002; (**h**) Ex. 635, *t* = 65.18, df = 2, *p* = 0.0002; (**i**) Ex. 488, *t* = 13.77, df = 2, *p* = 0.0052; (**i**) Ex. 561, *t* = 5.739, df = 2, *p* = 0.0290; (**i**) Ex. 635, *t* = 6.002, df = 2, *p* = 0.0267. *$p < 0.05$ **$p < 0.01$, ***$p < 0.001$. (mouse = 2 M, 1 F; marmoset= 2 M, 1 F; human = 3 F). Source data are provided as a Source Data file.

disruption of the lysosomal degradation pathway in microglia was shown to result in increased lipofuscin accumulation[19]. In addition, it is known that cell division is a primary way cells can reduce lipofuscin burden[13,14,22]. That is, as cells divide, lipofuscin is diluted. As microglia are actively dividing in the young brain, any lipofuscin and, therefore lipo-AF, may be diluted. In contrast, ongoing engulfment of cellular material in the adult brain, even at a low level, combined with decreased cell division may lead to accumulation of lipofuscin in microglia and its related AF. It is also important to note our observation that, while microglia have a high lipo-AF burden in the aged brain, the majority of lipo-AF that we detected was observed in neurons with aging. This neuronal accumulation of lipo-AF with age has been speculated to result from age-related decline in autophagy, lipophagy, and/or proteostasis[12].

Besides identifying age and region-dependent aspects of lipofuscin biology, we also show how lipofuscin can confound microglial synapse engulfment analyses by light microscopy. By measuring fluorescence intensity of engulfed/internalized anti-VGluT2 signal within microglia and lipo-AF at P90, we found these signal intensities to be equivalent. Similar results were obtained by immunostaining with another presynaptic marker anti-VGluT1. In contrast, immunostaining fluorescent intensity achieved with the postsynaptic marker anti-Homer1 was significantly higher than lipo-AF signal. This suggests that this postsynaptic signal may reflect bona fide engulfment

events in the adult brain. Also, it may be that postsynaptic material, which is enriched in molecular scaffolds, takes longer to digest in the lysosome and allows for more robust detection with antibodies. We sought to further explore the potential confound of lipo-AF on engulfment of presynaptic material and measured the area of anti-VGluT2 signal within microglia before and after photobleaching tissue, which we found to eliminate lipo-AF. In P90 brain, apparent engulfed anti-VGluT2 signal within microglia was no longer observed after photobleaching, demonstrating that detection of this apparent engulfed material was, indeed, from the lipo-AF signal. In contrast, in the P5 brain, anti-VGluT2 immunofluorescence signal was still observed within microglia after photobleaching. This latter experiment demonstrated that lipo-AF is less problematic for assessing synaptic engulfment in neonate tissue. Taken together, caution should be used when interpreting microglial engulfment, particularly in adult mouse brain. Of note, while we did most of our analyses with the 561 nm laser (BP 629/62), we detected lipo-AF within microglia with other laser lines, laser powers, and microscopes. As a result, this lipo-AF signal is likely a more universal issue across imaging set-ups. Photobleaching samples before immunostaining and including secondary-only controls to evaluate the amount of lipofuscin-derived AF or other sources of AF in tissue sections are the best practice.

As the last part of our study, we compared different lipo-AF quenching strategies and derived a cost-effective protocol for labs to

adopt going forward. We first found that quenching lipo-AF with the commercially available reagent TrueBlack Plus™ for 15 min post-immunostaining was sufficient to eliminate the AF signal. However, we also found a significant quenching of the immunostained fluorophore signal. It is possible to perform this prior to immunostaining to prevent this quenching, but detergent is not compatible with this reagent and will reduce the ability of antibodies to penetrate tissue for immunostaining. The second method consisted of incubating sections in a commercially available device that administers light to a sample for an extended period. Using this device, we found that photobleaching brain tissue sections for 12–24 h significantly reduced the AF signal without compromising fluorophore signal intensity. Finally, based on previous publications[21,29,30] and this commercial photobleacher, we designed a LED-based photobleaching system that had comparable ability to rid of AF in tissues. This homemade device is relatively inexpensive and simple to make, and it can easily be adopted across laboratories and used for multiple different types of tissue samples. However, besides this device and photobleaching samples prior to confocal light microscopy, assessment of synapse engulfment by microglia with multiple methods is a good practice. For example, this could include electron microscopy or tagging synaptic proteins with a biotin conjugate and assessing microglia lysosome content biochemically. These methods would also avoid the confounds of lipo-AF signal by light microscopy.

In summary, as more and more studies are realizing the impact of microglial engulfment mechanisms on neural circuit structure and function in health and disease[1–4], it is critical to perform experiments to assess engulfment of cellular and protein substrates by microglia to the highest rigor. The protocols we provide ensure that microglial engulfment confocal imaging assays are not confounded by AF. Importantly, our protocol and device are cost-effective, relatively simple to build, and we show they can be used for eliminating AF signal in mouse, non-human primate, and human tissue samples.

## Methods

All mouse data was acquired under the guidelines of the University of Massachusetts Chan Medical School Animal Care and Use Committees (IACUC) and NIH guidelines for ethics and proper use of animal welfare under protocol #PROTO202000113. Animals were housed in an AAALAC accredited barrier facility in individually ventilated cage racks. Irradiated chow (LabDiet 5P76) and chlorinated reverse osmosis purified water were provided to *ad libitum* and animals were housed on corn cob bedding and provided enrichment. Holding room light cycles were 12:12 h light-dark, room temperature and relative humidity (RH) were kept at 20–26 °C and 30–70% RH respectively. All marmosets were housed and handled with the approval of the NINDS/NIDCD/NCCIH Animal Care and Use Committee (ACUC) protocol #1308 and American Association for Laboratory Science (AALAS) guide. Marmosets were housed under conventional ventilation conditions on a 12 h light-dark cycle with *ad libitum* access to food and water. Room temperature was between 18 and 29 °C and humidity was between 30 and 70%. Human samples were acquired under the guidelines of the Translational Neuroradiology Section at the NIH/NINDS from a deidentified brain bank.

### Animals

Male and female wildtype C57Bl/6J mice (stock #000664) and Cx3cr1GFP/GFP mice (stock #005582) were obtained from Jackson Laboratories (Bar Harbor, ME). Adult common marmosets (*Callithrix jacchus*), both male and females between 11 and 13 years old, were obtained from the marmoset tissue library of translational neuroradiology section (TNS) at the NINDS. Sex was not factored into statistical analyses as sex was evenly distributed among data points. All animal experiments were performed in accordance with Animal Care and Use Committees (IACUC) and under NIH guidelines for proper animal welfare.

### Human samples

Collection of human multiple sclerosis (MS) postmortem brain tissue was performed after obtaining informed consent for collection and were obtained from the Translational Neuroradiology Section at the NIH/NINDS. Samples analyzed in the current study were collected from the insular/parietal neocortex and prefrontal cortex of 3 women with multiple sclerosis (MS) with ages ranging from 60 to 77 years. This was restricted to woman given the limited availability of tissue and the prevalence of MS towards females.

### Immunostaining

Mice were anesthetized and transcardially perfused with 0.1 M phosphate buffer (PB) followed by 4% paraformaldehyde (PFA) (Electron Microscopy Services 15710)/0.1 M PB. Brains were post-fixed at 4 °C in PFA overnight, equilibrated in 30% sucrose/0.1 M PB and then embedded in a 2:1 mixture of 30% sucrose/0.1 M PB and O.C.T. compound (ThermoFisher Scientific Waltham, MA, USA). To ensure methods were of global use, sections were immunostained on slides or floating. A cryostat was used to cut either 10–16 μm coronal brain sections on slides (microglial lipo-AF analysis across development and in 5XFAD mice) or 40 μm floating sections in 0.1 M PB (lipo-AF intensity, quenching and synaptic engulfment analysis). Subsequent sections were blocked and permeabilized at room temperature for 1 h in blocking solution (10% normal goat serum/0.1 M PB containing 0.3% Triton-X 100) followed by overnight incubation with primary antibodies at ambient room temperature. Primary antibodies included: Rat mAb anti-CD68 (Abcam, ab955; 1:200), rabbit pAb anti-IBA1 (Wako Chemicals, 019-19741; 1:500), chicken mAb anti-IBA1 (Synaptic Systems, 234009; 1:500), rabbit pAb anti-P2RY12 (Anaspec, 55043 A; 1:2000), guinea pig pAb anti-VGluT1 (Millipore, Ab5905; 1:1000), rabbit pAb anti-Homer1 (Synaptic Systems, 160003; 1:500), rabbit anti-Lyve1 (Abcam, ab14917; 1:200), mouse anti-NeuN (Millipore, MAB377; 1:200), rabbit anti-GFP (Millipore, MAB3080p; 1:1000) and guinea pig pAb anti-VGluT2 (Millipore, Ab2251-I; 1:1000). The following day, sections were washed 3×5 min with 0.1 M PB and incubated with the appropriate Alexa-fluorophore-conjugated secondary antibodies all diluted at 1:1000, including goat anti-chicken IgY (H + L) Alexa-Fluor 488 (Life Technologies Scientific; A11039), goat anti-mouse IgG(H + L) Alexa-Fluor 488 (Life Technologies; A11029), goat anti-rabbit IgG (H + L) Alexa-Fluor 488 (Life Technologies; A11034), goat anti-guinea pig IgG (H + L) Alexa-Fluor 488 (Life Technologies; A11073), goat anti-rabbit IgG (H + L) Alexa-Fluor 594 (Life Technologies; A11012), goat anti-guinea pig IgG (H + L) Alexa-Fluor 594 (Life Technologies; A11076), goat anti-rabbit IgG (H + L) Alexa-Fluor 647 (Life Technologies; A21245), goat anti-guinea pig IgG (H + L) Alexa-Fluor 647 (Life Technologies; A21450), goat anti-rat IgG (H + L) Alexa-Fluor 647 (Life Technologies; A21247) for 2 h at room temperature. Slides and floating sections were washed 3 × 10 min with 0.1 M PB. Floating sections were then mounted on slides. All subsequent slides were air dried and cover glass (ThermoFisher; 12-544-DP) was mounted with Vectashield containing DAPI (Vector laboratories, Burlingame, CA, USA) or with CFM-3 (Citifluor, Hatfield, PA, USA) for chemical quenching experiments.

### Confocal imaging

Mounted brain sections were imaged using Zen Blue acquisition software (Zeiss; Oberkochen, Germany) on a Zeiss Observer Spinning Disk confocal microscope equipped with diode lasers 405 nm/50 mW, 488 nm/50 mW, 561 nm/50 mW, and 638 nm/75 mW, and with 450/50 (blue), 525/50 (green), 629/62 (red) and 690/50 (far-red) BP emission filter sets, respectively. For Supplementary Fig. 1a, b, images were taken on a Zeiss LSM 700 with 405, 488, 555 and 639 nm lasers with bandpass filters 450–490 and 640/30. For Supplementary Fig. 1c, d, images were taken on a Leica Lightening SP8 Inverted Laser Scanning Confocal microscope, with 405, 488, 552 and 638 nm lasers with bandpass filters 445/50, 525/50, 605/70 and 690/50 using LAS X

software. For most experiments, 6–12 (AF) or 3 (anti-VGluT2 immunostained sections) 40x fields of view were randomly chosen within the somatosensory and neighboring visual and auditory cortices an z-stacks were acquired at 0.31 μm spacing. For AF, anti-VGluT2, anti-Homer1, anti-VGluT1, anti-CD68 and anti-P2RY12 intensity measurements, 2–4 63x fields of view were randomly and z-stacks were then acquired at 0.27 μm spacing. For all imaging experiments, identical settings were used to acquire images from all samples within one experiment.

### Image quantification: volume measurements

To quantify the volume of lipo-AF or anti-VgluT2 within individual microglia, images were first pre-processed and blinded using a custom macro in ImageJ (NIH, version 1.53c). This includes application of a uniform background subtraction by channel, despeckling and conversion to TIF before blinding. Imaris v9 (Bitplane) was then used to create a 3D surface rendering of single microglia cells. A 3D surface rendering of masked anti-CD68+ signal from each cell was also created and was used to generate a 3D surface rendering of CD68-masked AF+ or CD68-masked VGluT2+ material. Lipo-AF burden or VGluT2 engulfment within microglia was calculated by dividing the volume of CD68-masked lipo-AF or VGluT2 by the volume of the individual microglial cell. Colocalization in Fig. 4a–c was calculated by using colocalization of objects in IMARIS 10.0.0.

To quantify lipo-AF volume within a field of view, ImageJ was used to create a binary mask of lipo-AF and the immunostained signal (anti-CD68, anti-NeuN, or anti-Lyve-1) across z-planes. Using Image calculator function, a binary mask of total lipo-AF or colocalized lipo-AF with immunofluorescence signal from anti-CD68, anti-NeuN, or anti-Lyve-1 was generated. The voxel volumes were then calculated using MorphoLibJ plug-in. The percentage of non-microglial lipo-AF was then calculated by subtracting the CD68-masked, lipo-AF volume from total lipo-AF and dividing this number by to total lipo-AF. The percentage of neuronal or border-associated macrophage lipo-AF was calculated by dividing NeuN-masked lipo-AF or Lyve-1-masked lipo-AF by total lipo-AF, respectively.

To assess lipo-AF volume in 5XFAD tissue, images were pre-processed (background subtraction and despeckle) using a custom macro in ImageJ (NIH, version 1.53c). ImageJ was then used to create a binary mask of ThioS, IBA1, lipo-AF and CD68 signal across several z-slices. A 25 μm radius circumference region of interest (ROI) was placed at the center of ThioS+ plaques and these regions were either subtracted from the original binary mask to generate binary masks of non-plaque regions or duplicated if the circumference did not overlap with neighboring plaques to generate single plaque regions. Using the Image calculator function, binary masks of total ThioS+, total lipo-AF+ or IBA1-/CD68-masked lipo-AF+ material was generated, and their voxel volumes were calculated using MorphoLibJ plug-in. Percentage ratio of non-microglial lipo-AF was calculated by subtracting the CD68-masked lipo-AF volume from total-lipo-AF and divided by total lipo-AF. Lipo-AF burden within microglia was calculated by dividing the volume of IBA1-masked lipo-AF by the volume of IBA1+ microglia.

### Image quantification: fluorescence intensity measurements

To quantify lipo-AF signal intensity within a field, the find maxima function was used in Image J to identify lipo-AF puncta, a 2 μm line was drawn at its center and the Plot profile function was used to measure and plot gray values across each line drawn. Area under the curve (AUC) was then calculated using GraphPad Prism (La Jolla, CA). To compare synaptic and lipo-AF intensity analyses were performed on single z-planes using ImageJ (NIH, version 1.53c). Briefly, for sections stained with synaptic markers (anti-VGluT2, anti-VGluT1, or anti-Homer) and anti-IBA1, square regions of interests (ROIs) of equal size were drawn at the center of anti-IBA1 immunolabelled microglial somas containing synaptic marker positive signal. Within these ROIs,

smaller circular ROIs of equal size were placed at the center of synaptic marker immunolabelled puncta outside and inside anti-IBA1 immunolabelled microglia. Three background circular ROIs of the same size were also selected for each field of view within the single z-plane. The raw integrated density of pixels within each circular ROI was measured. To quantify lipo-AF intensity, unstained sections were imaged and subsequently ROIs of equal size were drawn at the center of single z-planes containing multiple lipo-AF puncta. Similar to immunostained sections, circular ROIs of equal size were then placed at the center of AF. Three background ROIs were also selected. For all images (immunostained with VGluT2 or unstained to measure AF), the raw integrated density of pixels within each circular ROI was measured and the background ROI pixel intensity value was averaged. Last, for each synaptic or AF+ puncta intensity measurement, the average background was subtracted prior to statistical comparison. To quantify the fluorescence intensity of anti-P2Y12R and anti-CD68 signal, ImageJ (NIH, version 1.53c) was used to create a binary mask across z-slices and saved to ROI manager. These masks were then used on the original z-slices and the fluorescence intensity was measured within each mask.

### Chemical quenching

After staining, floating sections or tissue-mounted slides were incubated with TrueBlack® Plus Lipofuscin Autofluorescence Quencher (Biotium, Fremont, CA, USA) in 1X phosphate buffer saline (PBS) for 15 min at room temperature with rocking. Following 3 × 5 min washes with PBS 1X, sections were mounted with CFM-3 (Citifluor, Hatfield, PA, USA).

### Photobleaching

Before incubating in blocking solution, floating sections or tissue-mounted slides were placed in 0.1 M PB and incubated in the MERSCOPE Photobleacher (Vizgen, Cambridge, MA, USA) for 12–24 h. Photobleached samples were then incubated in blocking solution and immunostained as described above. The lab-made photobleacher was created using the HIGROW 36 W LED Plant Grow Light Bulb with 18x2W 450–460 nm Blue LEDs (HIGROW), aluminum foil (Reynolds), HILAMPCORDLX2V1 15 ft Extension Hanging Lantern E26 Socket (Simple Deluxe) and NDS 107BC Standard Meter Box. The LED Bulb was hung ~3 inches above floating sections or tissue-mounted slides placed in 0.1 M PB inside aluminum foil coated chamber and incubated for 24 h. Photobleached samples were then incubated in blocking solution and immunostained as described above.

### Statistical analysis

Results are presented as either mean or mean ± standard error (SEM). For normally distributed data, analyses included two-tailed, unpaired Students $t$ test when comparing 2 conditions or one-way ANOVA followed by Tukey's or Dunnet's post hoc analysis or two-way ANOVA followed by Sidak's post hoc analyses (indicated in figure legends) using GraphPad Prism (La Jolla, CA). Imaging in 1a and 3a were repeated three independent times while 2a and 2d were repeated four independent times. For data that were not normally distributed, a Mann-Whitney test was used. Values of *$P < 0.05$, **$P < 0.01$, ***$P < 0.001$, ****$P < 0.0001$ were considered statistically significant. A single P60 timepoint was excluded from this study using Grubbs' Outlier test for Fig. 4a–c.

### Reporting summary

Further information on research design is available in the Nature Portfolio Reporting Summary linked to this article.

## Data availability

All source data are provided with this paper. All other data that support the findings of this study are available from the corresponding author

upon reasonable request. All materials and correspondences should be addressed to Dr. Dorothy P. Schafer dorothy.schafer@umassmed.edu. Source data are provided with this paper.

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

## Acknowledgements

We thank Dr. Christina Baer (UMass Chan) for assistance with microscopy and design of the homemade photobleacher. We thank Joan Ohayon, CRNP, for coordinating MS autopsies. This work was supported by NIMH-R01MH113743 (D.P.S.), NINDS-R01NS117533 (D.P.S.), NIA-RF1AG068281 (D.P.S.), NIH-T32AI132152 (J.M.S.), the National Multiple Sclerosis Society TA-2203-39387 (J.-P.L), the Intramural Research Program of NINDS (D.S.R.), and the Dr. Miriam and Sheldon G. Adelson Medical Research Foundation (D.P.S. and D.S.R.).

## Author contributions

J.M.S., F.M.L. and D.P.S. designed the study and wrote the manuscript. J.M.S., F.M.L. and D.P.S. performed and analyzed the experiments. D.S.R. supervised collection and processing of marmoset and human brain tissue by J.-P.L. and K.H. All authors edited and revised the manuscript.

## Competing interests

The authors declare no competing interests.
