## [Peer Review File · Nature Communications]

Lipofuscin-like autofluorescence within microglia and its impact on studying microglial engulfmentREVIEWER COMMENTS

Reviewer #1 (Remarks to the Author):

In this manuscript entitled 'Lipofuscin-like autofluorescence within microglia and its impact on studying microglial engulfment', Stillman and colleagues aimed to develop a method to quench autofluorescence (AF) signal in microglia that could potentially confound downstream analyses. The authors first characterized AF in the somatosensory cortex at different timepoints and found that microglia exhibit more AF over time. Further, the authors hypothesize lipofuscin within lysosomes to be the main source of AF, that could potentially interfere with popular histological analyses within microglia such as the synaptic engulfment assay, introduced by the senior author during her postdoctoral training at the Stevens lab (Schafer et al., 2014). Indeed, data demonstrated in the current manuscript suggests that lipofuscin- AF could potentially be misinterpreted as false-positive synaptic immunoreactivity when analysed using the yellow/green (561nm) and far-red laser, but not UV laser. Of interest, the authors did not find significant increase of AF in microglia in the 5xFAD model. The authors finally adapted their model, based on prior quenching of any background using either chemical or light-based bleaching, to mouse, marmoset and human brain tissue.

The manuscript could be of potential interest to the microglia community because it offers a protocol that could improve reproducibility of immunofluorescence-based experiments. However, experimental shortcomings and the general lack of originality limits my enthusiasm for this manuscript. The authors introduce a model for an artefact that one can potentially overcome by using the appropriate controls and careful considerations, for example only secondary antibody incubation control, adjustment for laser power etc. In fact, many research laboratories do not have access to Merscope equipment suggested by the authors. Evidence provided by the authors that AF could interfere with downstream analyses (engulfment assay, but are there any others?) and that signal quenching could improve data interpretation, should be more convincing and robust. Please find detailed points and suggestions below:

Major points/suggestions:

1) In Fig. 4, the authors demonstrate that synapse immunoreactivity within microglia is comparable to the percentage of AF within microglia of an adjacent brain slice, suggesting that synapse immunoreactivity could potentially be misinterpreted due to AF. This is not very convincing, and the authors should provide more rationale to motivate different labs to adapt the suggested protocol. For example, could the authors design an experiment where colocalization is measured between synapses and AF, by immunostaining for synapses using fluorescent dyes in the red spectra (for example, alexa 647) and AF excited at 488nm? There does not seem to be a significant increase in autofluorescence using the red (638 nm) laser compared to the UV laser (Fig 1b and c). I think the obtained data could be more informative and a better measure for potential false-positive synapse staining caused by AF. In summary, if the authors could demonstrate that AF could lead to false-positive synapse immunoreactivity in a more convincing way, this would greatly contribute to the significance of the

manuscript.

2) The authors demonstrate that the quenching protocol does not compromise P2Y12 immunostaining. Could the authors perform additional experiments to demonstrate the preservation of fluorescent signal of microglial markers of interest within the marmoset and human tissue? It would also be important to demonstrate that this protocol does not compromise lysosomal CD68+ and LAMP1+ immunostaining when considering photobleaching prior to synaptic engulfment assay. In this regard, CD68 staining is demonstrated in figure 4D and 4E after treatment, so I assume signal will not be quenched?

3) It would be of great interest to characterise whether there is region-specific appearance of AF+ microglia. For example, the authors mention a subset of AF+ microglia (line 88) that appears during postnatal development. In contrast, in the 5xFAD model, there seems to be no spatial heterogeneity in AF+ microglia located either close or further away from plaques. Could the authors provide more experimental information on where exactly AF+ microglia are located? For example, is there difference between white and grey matter regions? Further, it would be informative to provide an overview of AF+ microglia abundance in regions that are relevant to microglia-mediated synaptic engulfment (for example, barrel cortex, retinogeniculate system,...). Finally, it would be of interest for the manuscript if the authors can speculate more on AF+ microglia heterogeneity in the discussion part. For example, do the authors speculate that certain microglia subpopulations are more prone to AF accumulation? In line, but probably not the scope of this manuscript, it would be interesting to probe whether the AF+ microglia acquire a distinct functional phenotype due to potential lipofuscin accumulation.

Minor points:

-I assume background signal and AF is in part determined by laser power, so it would be useful if the authors could mention the laser power settings in the materials and method part.

-Please provide dot plots or show individual datapoints in each figure for transparency. Please clearly indicate what one data point represents in each figure legend (1 datapoint= one microglia, one region of interest, 1 animal...).

-With regards to the supplementary figure, could the authors provide representative lysosomal stainings for each time point assessed in figure 1? It will be of interest to see whether the ratio AF volume to lysosomes/ other cellular compartments varies over the different time points.

-Related to my previous point, it seems that microglia are not the main source of the AF at later time points, an interesting observation mentioned but not further investigated by the authors. For example, by looking at figure 1e-24mo, it seems to be that microglia represent only a small proportion of the AF+ cells. Could the authors further speculate on this in the discussion?

-I assume the protocol quenches any endogenous fluorescence (for example GFP or TdTomato, often used in microglia-based reporter mice), but would it be possible to retain the signal by staining for anti-GFP or anti-RFP, for example?

-It is not clear from the main text whether the synaptic engulfment assay represents VGLUT2 staining within IBA1+ cells or CD68+ IBA1+ lysosomes. Could the authors make this more clear in the text?

-Line 72-73, please provide references for the following sentence "We focused our imaging in the somatosensory cortex and neighboring visual and auditory cortices as this was a large region that could be easily identified across ages and is known to undergo neurodegeneration REF"

-Line 85: Liop-AF should be Lipo-AF

-Can the authors touch on AF within border-associated macrophages? These cells are receiving increased attention recently, so it would be of interest to demonstrate whether BAM are a potential source of lipofuscin-AF or not.

Reviewer #2 (Remarks to the Author):

In this manuscript, the authors examine a key aspect of microglial biology – phagocytic engulfment of cellular material – and highlight how light microscopic study of this microglial function can be confounded by autofluorescent (AF) material within these cells. As the authors themselves note, a growing number of studies point to microglial engulfment of synaptic material, extracellular matrix, myelin, dying cells, and pathological protein aggregates as playing a key role in shaping circuit function during development, health and disease. Many of these data rely on light microscopy and colocalization of potentially engulfed material with microglial lysosomes. Yet, the AF material in microglia is also localized to lysosomes, indicating that this approach cannot unequivocally confirm microglial engulfment.

Here, the authors analyze at what age AF signals appear in cortical microglia and show that TrueBlack or photobleaching can be used to remove these AF signals. They examine whether microglia near to Abeta plaques have exacerbated AF accumulation, and find, surprisingly, that they do not. Most importantly, they examine how incorporation of photobleaching approaches impacts quantification of microglial synapse engulfment in early postnatal and young adult mice. Their results indicate that much of the putative engulfment observed in young adult mice is likely a “false-positive” result due to AF accumulation.

On the whole the manuscript is well-written, the figures clearly presented, and the topic of accurate analysis of microglial engulfment is an important one for the field. However, the analyses presented are somewhat superficial and there are key concerns about the novelty as other studies have already detailed how to use photobleaching and other approaches to remove AF signals from brain tissue.

Major Concerns

1) This reviewer’s main concern is linked to several other recent publications (most of which the authors cite) that have provided detailed methods for detecting and removing AF from light microscopic analysis of rodent, NHP, and human brain tissue PMID: 31790435, PMID: 31379520, PMID: 28892031, PMID: 23994358, PMID: 35361108. This includes both chemical methods like True Black and photobleaching approaches. Hence, it feels like an overstatement to say that the current manuscript provides the field with a “new protocol” (line 189) that can be used to get rid of AF. Moreover, use of these approaches has already been demonstrated in human tissue PMID: 28892031. One of the above publications (PMID:

35361108) explores AF specifically in microglia and analyzes how this impacts flow cytometry and light microscopy in fixed tissue, acute brain sections, and in vivo imaging. They show which flow cytometry fluorophores are least susceptible to AF interference and that True Black can be used to remove AF signals from fixed tissue. Further, they point out the implications of these AF granules for complicating interpretation of microglial engulfment in their discussion. Hence, the novelty of the present study is reduced somewhat.

2) The direct examination of how photobleaching AF affects quantification of microglial synapse engulfment in cortex is novel. But this is explored in a relatively superficial manner. Does the concentration of VGlut2 antibody (and secondary antibody) used affect the degree to which AF granules are misinterpreted as microglial engulfment of synapses? Presumably the dimmer the antibody-labeled signal, the more one will need to turn up laser intensity / detector gain and the more AF signals will create problems. Similarly, performance of individual antibodies varies widely – is the degree to which AF introduces spurious engulfment results similar for synaptic terminals labeled with antibodies against VGlut1, synaptophysin, bassoon or similar? Are spurious engulfment results avoided even without photobleaching if 405 secondaries are used rather than red? How much are spurious engulfment results affected by postfixation length or how long brain sections have been in storage (and whether they were protected from light during storage and staining)? How much is the degree of spurious engulfment results affected by the imaging system? It would be of interest to know whether microglial AF is similarly problematic if tissue sections from the same mice, stained in the same way (+/- photobleaching) are imaged on systems with distinct optics, laser type and age, and detector type. Essentially – it feels like these authors would be in a unique position, given their expertise, to provide the field with more information about how prevalent spurious engulfment results may be and what are the parameters that most increase the risk of engulfment analyses being contaminated by AF granules. This is the type of information that would truly elevate this study above what has already been done in the field.

3) If the main point of the paper is that AF material could be mistaken for engulfed material, it seems like they need to show this. Currently, they only show that photobleaching alters the amount of VGlut2 signal detected as being inside microglia at P90. Can they put the VGlut2 antibody in far red and leave the red channel open to show how many instances of “engulfed” VGlut2 puncta have an AF signal in the red channel? Of course, VGlut2 synaptic terminals could have been engulfed and trafficked to lysosomes that also have AF material.....but, together with the photobleaching experiments, this would directly demonstrate how AF signals could be mistaken for engulfment signals.

4) TrueBlack can also be applied during blocking and prior to antibody staining. This would likely avoid the loss in P2RY12 signal that the authors observed. If this were true, it would greatly strengthen the manuscript to be able to compare removal of AF via photobleaching and removal of AF via TrueBlack and show whether these approaches are equally effective in rendering microglial VGlut2 engulfment analysis more robust (at least in rodent tissue). Did the authors try TrueBlack application prior to antibody staining?

5) The observation that microglia near to A-beta plaques potentially have less AF is surprising and also a novel observation. However, this, too, is explored only in a very cursory way. What % of plaques have microglia with little or no AF near them? How is this effect impacted by the size of the plaque? Is this

effect only noticed in microglia that are immediately adjacent to / potentially contacting plaques? What is the status of the lysosomes in the plaque-adjacent microglia with little or no AF?

Minor Concerns

1) Throughout the manuscript, the images appear to have had quite a bit of contrast introduced. The methods mention that images were “pre-processed” before analysis but what this involves isn’t described. Some of the images also appear quite pixelated.

2) Fig 1F, 2C, 3C, 3J, graphs seem redundant with the individual graphs that accompany them and that have the statistics and this causes some confusion (the reader is trying to determine if something different is being shown or just the same thing in different formats). This reviewer would also recommend giving these graphs different titles “negligible AF burden (0-0.1% cell vol)” “small AF burden (0.1-1% cell vol)” “moderate AF burden (1-2% cell vol)” “severe AF burden (>2.0% cell vol)” or similar. “% AF within microglia” at first seems like it’s referring to how much AF is in the microglia versus how much is outside of the microglia.

3) In figure 4F, the authors should show the raw data – not just the ratio of detected engulfment with or without photobleaching. This would give greater transparency about the results and structure of the data.

4) It would be helpful to show some lower magnification images of VGlut2 staining (and possibly staining for some other punctate markers – e.g. Vglut1, synaptophysin, aggrecan) in P5 and P90 with and without photobleaching. This would give a better sense of how much the overall landscape of labeling with these markers is altered by photobleaching and removal of AF.

5) The manuscript seems to be overselling the finding that AF shows up first in microglia before other cells. To claim this more strongly, they would ideally look at the timecourse of AF increasing in microglia and other cells in all the wavelengths, not just red, and with other ages between P90 and 24mo.

6) In the discussion it might be nice to focus less on the implications of lipofuscin for microglial biology (since this isn’t a primary focus of the study). Instead, they could include discussion of strategies for accurate detection of AF (such as spectral unmixing). Is this a strategy that researchers could use in their systems to determine how much AF might be a concern for engulfment analyses? They could also discuss complementary approaches for detecting microglial engulfment of materials, such as EM analysis or immunohistochemical (non-fluorescent) imaging based methods. It would also be lovely to see a discussion of tools and approaches not yet available in the field to study microglial engulfment more robustly. TurboID targeted to microglial lysosomes to allow tagging and subsequent protein level analysis of material that is trafficked to microglial lysosomes?

Reviewer #3 (Remarks to the Author):

This work addresses an important gap in knowledge in microglial biology. The authors did an excellent job identifying a novel method to decrease and deplete immunostained tissue of auto fluorescent lipo-like AF's within microglia. This technique is even more valuable given their data suggesting lipo-AF's are confounding other typical engulfment contents of microglia, especially during crucial periods of synaptic pruning earlier in development. Overall, this work and technique is important to better understand the substrates microglia are clearing without any auto fluorescent interference. Implementing this technique in labs that study phagocytosis. would be crucial as to not misinterpret the engulfment contents of microglia in general, across development, and in instances of neurodegenerative disease. I have only a few minor concerns and questions outlined below.

Overall minor concerns:

General:

Please report/display individual data points overlaid onto bar graphs in all %AF graphs like those in Figure 1f-1k. This is important given interest in heterogeneity in microglia in many brain regions, beyond simple transparency in data reporting.

Figure 1:

- In the instances where lipo-AF was not co-localized within CD68+ lysosomes or Iba1+ microglia (Figure 1k), Is it possible that the older microglia are reaching a maximum phagocytic capacity?
- Is there another subset of microglia that are not Iba1+, or maybe another glial or macrophage population that are able to engulf the lipo-AF in the older animals? Or are the microglia themselves potentially undergoing cell death or apoptosis and the accumulations of lipo-AF that are left behind following this? The amount of unengulfed excess lipo-AF at the 24m age is very interesting, and worth exploring whether another non-Iba1+ cell type is engulfing the lipo-AF outside of the microglia. Could you use other macrophage/microglia markers to label additional cells that could be engulfing these?
- In Figure 1K, was there a significant difference between p5 and p15 & p30? It is interesting that the lipo-AF volume appears higher in earlier timepoints and then tapers, similar to crucial timepoints of synaptic pruning in the developing CNS. Maybe this could also be younger microglia reaching phagocytic capacity with synaptic debris and they cannot uptake as much of the lipo-AF?

Figure 2:

No comments

Figure 3:

No comments, the comparison of the two distinct techniques and their strengths and weaknesses is thoroughly addressed in the figure and discussion sections.

Figure 4:

- This is a super exciting technique and finding that lipo-AF can impact the interpretation of microglial contents. Is this something that could be (or should be) used for tissue earlier in development? Or does the photobleaching have an age threshold in which it works best between P5 and P90?

Figure 5:

No comments, very important cross-species comparison.

We thank the reviewers for their careful and thorough review of the manuscript, which has only improved as a result. Any changes are highlighted in red text in the revised manuscript.

Reviewer #1 (Remarks to the Author):

In this manuscript entitled ‘Lipofuscin-like autofluorescence within microglia and its impact on studying microglial engulfment’, Stillman and colleagues aimed to develop a method to quench autofluorescence (AF) signal in microglia that could potentially confound downstream analyses. The authors first characterized AF in the somatosensory cortex at different timepoints and found that microglia exhibit more AF over time. Further, the authors hypothesize lipofuscin within lysosomes to be the main source of AF, that could potentially interfere with popular histological analyses within microglia such as the synaptic engulfment assay, introduced by the senior author during her postdoctoral training at the Stevens lab (Schafer et al., 2014). Indeed, data demonstrated in the current manuscript suggests that lipofuscin- AF could potentially be misinterpreted as false-positive synaptic immunoreactivity when analysed using the yellow/green (561nm) and far-red laser, but not UV laser. Of interest, the authors did not find significant increase of AF in microglia in the 5xFAD model. The authors finally adapted their model, based on prior quenching of any background using either chemical or light-based bleaching, to mouse, marmoset and human brain tissue.

The manuscript could be of potential interest to the microglia community because it offers a protocol that could improve reproducibility of immunofluorescence-based experiments. However, experimental shortcomings and the general lack of originality limits my enthusiasm for this manuscript. The authors introduce a model for an artefact that one can potentially overcome by using the appropriate controls and careful considerations, for example only secondary antibody incubation control, adjustment for laser power etc. In fact, many research laboratories do not have access to Merscope equipment suggested by the authors. Evidence provided by the authors that AF could interfere with downstream analyses (engulfment assay, but are there any others?) and that signal quenching could improve data interpretation, should be more convincing and robust. Please find detailed points and suggestions below:

Response: We thank the reviewer for their careful review of our manuscript and for appreciating the value to the community. We have done our best to address the reviewer’s experimental critique below and our manuscript has improved as a result. In regards to originality, it is true that microglial lipofuscin accumulation is known and that there are existing protocols for quenching autofluorescence. We are of the opinion that the importance of this manuscript is more in its “cautionary tale” to the field that this lipo-AF can interfere with synapse engulfment, as well as engulfment of other substrates. These sorts of studies are so critical to move the field forward in an arena where it will be read broadly. We also think the photobleaching protocol is extremely useful to labs working inside and outside of the brain and we appreciate the reviewer’s concern about the MERSCOPE photobleacher’s accessibility. We, therefore, now provide a protocol to build your own inexpensive, simple photobleaching device (**New Fig. 6 and new Supplementary Fig. 4**).

Major points/suggestions:

1) In Fig. 4, the authors demonstrate that synapse immunoreactivity within microglia is comparable to

the percentage of AF within microglia of an adjacent brain slice, suggesting that synapse immunoreactivity could potentially be misinterpreted due to AF. This is not very convincing, and the authors should provide more rationale to motivate different labs to adapt the suggested protocol. For example, could the authors design an experiment where colocalization is measured between synapses and AF, by immunostaining for synapses using fluorescent dyes in the red spectra (for example, alexa 647) and AF excited at 488nm? There does not seem to be a significant increase in autofluorescence using the red (638 nm) laser compared to the UV laser (Fig 1b and c). I think the obtained data could be more informative and a better measure for potential false-positive synapse staining caused by AF. In summary, if the authors could demonstrate that AF could lead to false-positive synapse immunoreactivity in a more convincing way, this would greatly contribute to the significance of the manuscript.

Response: We now provide a more comprehensive assessment of colocalization of lipo-AF with VGLuT2 (**new Fig. 4a-c**). We also show in **Fig. 5k-n** that apparent engulfed VGLuT2 in adult animals is eliminated after photobleaching. This last experiment confirms that lipo-AF can be confused for anti-synaptic staining inside (but not outside) microglia. We further show analyses with other synaptic markers (**new Fig. 4d-i**)

2) The authors demonstrate that the quenching protocol does not compromise P2Y12 immunostaining. Could the authors perform additional experiments to demonstrate the preservation of fluorescent signal of microglial markers of interest within the marmoset and human tissue? It would also be important to demonstrate that this protocol does not compromise lysosomal CD68+ and LAMP1+ immunostaining when considering photobleaching prior to synaptic engulfment assay. In this regard, CD68 staining is demonstrated in figure 4D and 4E after treatment, so I assume signal will not be quenched?

Response: To address this concern, we have shown that anti-CD68 signal is not significantly reduced with our homemade photobleacher (**new Supplementary Fig. 5C**), anti-IBA1 does not quench with mouse, human MS and marmoset tissue (**new Supplementary Fig. 6d-k**), and anti-GFP signal does not significantly quench (**new Supplementary Fig. 6a-b**).

3) It would be of great interest to characterise whether there is region-specific appearance of AF+ microglia. For example, the authors mention a subset of AF+ microglia (line 88) that appears during postnatal development. In contrast, in the 5xFAD model, there seems to be no spatial heterogeneity in AF+ microglia located either close or further away from plaques. Could the authors provide more experimental information on where exactly AF+ microglia are located? For example, is there difference between white and grey matter regions? Further, it would be informative to provide an overview of AF+ microglia abundance in regions that are relevant to microglia-mediated synaptic engulfment (for example, barrel cortex, retinogeniculate system,...). Finally, it would be of interest for the manuscript if the authors can speculate more on AF+ microglia heterogeneity in the discussion part. For example, do the authors speculate that certain microglia subpopulations are more prone to AF accumulation? In line, but probably not the scope of this manuscript, it would be interesting to probe whether the AF+ microglia acquire a distinct functional phenotype due to potential lipofuscin accumulation.

Response: This is an excellent point. We now show that microglia AF accumulates first in white matter vs. gray matter (**new Fig. 2 and Supplementary Fig. 2**). However, in aged mice, AF accumulates throughout microglia in white and gray matter regions to similar levels. We further expand the discussion to speculate on why white matter microglia may accumulate lipo-AF sooner than gray matter.

Minor points:

-I assume background signal and AF is in part determined by laser power, so it would be useful if the authors could mention the laser power settings in the materials and method part.

Response: We now include a more detailed description of our microscope settings. We further add new data that lipo-AF can be detected at different laser powers with the same microscope (**new Fig. 1d-f**) and it can be detected with other microscopes (**new Supplementary Fig 1**).

-Please provide dot plots or show individual datapoints in each figure for transparency. Please clearly indicate what one data point represents in each figure legend (1 datapoint= one microglia, one region of interest, 1 animal...).

Response: These individual data points have now been added and we define the n's as animals (1 data point=1 animal).

-With regards to the supplementary figure, could the authors provide representative lysosomal stainings for each time point assessed in figure 1? It will be of interest to see whether the ratio AF volume to lysosomes/ other cellular compartments varies over the different time points.

Response: We now added representative CD68+ lysosomal stainings (**new Fig. 2a,d**) and quantify this staining for each time point (**new Supplementary Fig. 2c,e**).

-Related to my previous point, it seems that microglia are not the main source of the AF at later time points, an interesting observation mentioned but not further investigated by the authors. For example, by looking at figure 1e-24mo, it seems to be that microglia represent only a small proportion of the AF+ cells. Could the authors further speculate on this in the discussion?

Response: We agree. This is interesting. We emphasize that microglia still have a very large and significantly increased lipo-AF burden at older ages. However, other cells do start to accumulate lipo-AF with age. As microglia are smaller in number compared to other cell types, total lipo-AF outside microglia is higher at older ages. We now show that most of lipo-AF outside of microglia at older ages is from neurons (**new Supplementary Fig. 2h-i**). We add further discussion of this timeline of lipo-AF accumulation in microglia vs. other cells in the revised manuscript discussion.

-I assume the protocol quenches any endogenous fluorescence (for example GFP or TdTomato, often used in microglia-based reporter mice), but would it be possible to retain the signal by staining for anti-GFP or anti-RFP, for example?

Response: Yes, with antibody staining amplification, endogenous fluorescence can still be visualized after photobleaching. We added these data to the revised manuscript (**new Supplementary Fig. 5a-b**).

-It is not clear from the main text whether the synaptic engulfment assay represents VGLUT2 staining within IBA1+ cells or CD68+ IBA1+ lysosomes. Could the authors make this more clear in the text?

Response: This is engulfment assay represent VGluT2 within CD68+/IBA1+ lysosomes. We now clarify this in the text.

-Line 72-73, please provide references for the following sentence "We focused our imaging in the somatosensory cortex and neighboring visual and auditory cortices as this was a large region that could be easily identified across ages and is known to undergo neurodegeneration REF"

Response: Thank you for catching this. This sentence has now been omitted in the revised manuscript.

-Line 85: Liop-AF should be Lipo-AF

Response: This has been changed. Thank you for pointing this out.

-Can the authors touch on AF within border-associated macrophages? These cells are receiving increased attention recently, so it would be of interest to demonstrate whether BAM are a potential source of lipofuscin-AF or not.

Response: We have now added new data showing and quantifying AF within border-associated macrophages (**new Supplementary Fig. 2f-g**).

Reviewer #2 (Remarks to the Author):

In this manuscript, the authors examine a key aspect of microglial biology – phagocytic engulfment of cellular material – and highlight how light microscopic study of this microglial function can be confounded by autofluorescent (AF) material within these cells. As the authors themselves note, a growing number of studies point to microglial engulfment of synaptic material, extracellular matrix, myelin, dying cells, and pathological protein aggregates as playing a key role in shaping circuit function during development, health and disease. Many of these data rely on light microscopy and colocalization of potentially engulfed material with microglial lysosomes. Yet, the AF material in microglia is also localized to lysosomes, indicating that this approach cannot unequivocally confirm microglial engulfment.

Here, the authors analyze at what age AF signals appear in cortical microglia and show that TrueBlack or photobleaching can be used to remove these AF signals. They examine whether microglia near to Abeta plaques have exacerbated AF accumulation, and find, surprisingly, that they do not. Most importantly, they examine how incorporation of photobleaching approaches impacts quantification of microglial synapse engulfment in early postnatal and young adult mice. Their results indicate that much of the putative engulfment observed in young adult mice is likely a “false-positive” result due to AF accumulation.

On the whole the manuscript is well-written, the figures clearly presented, and the topic of accurate analysis of microglial engulfment is an important one for the field. However, the analyses presented are somewhat superficial and there are key concerns about the novelty as other studies have already detailed how to use photobleaching and other approaches to remove AF signals from brain tissue.

Response: We greatly appreciate the reviewer’s feedback and for appreciating the value of our study to the community. We have done our best to address the reviewer’s concerns below. In regards to novelty, it is true that microglial lipofuscin accumulation is known and that there are existing protocols for quenching autofluorescence. We are of the opinion that, the most important aspect of this manuscript, as the reviewer also points out, is more in its “cautionary tale” to the field. That is, this lipo-AF can interfere with synapse engulfment, as well as engulfment of other substrates. These sorts of studies are so critical to move the field forward and it is critical that they are published in an arena where it will be read broadly. Also, while photobleaching has been shown to rid of autofluorescence in brain tissues before, this has never been shown across species. We now include in the revised proposal, imaging of lipofuscin with different laser intensities (**new Fig. 1d-f**), different microscope set-ups (**new Supplementary Fig. 1**), and different synaptic antibodies (**new Fig. 4d-i**). Also, in the revised manuscript, we provide a protocol (**new Fig. 6 and Supplementary Fig. 4**) to build your own inexpensive photobleaching device with similar photobleaching efficacy as the MERFISH photobleacher.

Major Concerns

1) This reviewer’s main concern is linked to several other recent publications (most of which the authors cite) that have provided detailed methods for detecting and removing AF from light microscopic analysis of rodent, NHP, and human brain tissue PMID: 31790435, PMID: 31379520, PMID: 28892031, PMID: 23994358, PMID: 35361108. This includes both chemical methods like True Black and photobleaching

approaches. Hence, it feels like an overstatement to say that the current manuscript provides the field with a “new protocol” (line 189) that can be used to get rid of AF. Moreover, use of these approaches has already been demonstrated in human tissue PMID: 28892031. One of the above publications (PMID: 35361108) explores AF specifically in microglia and analyzes how this impacts flow cytometry and light microscopy in fixed tissue, acute brain sections, and in vivo imaging. They show which flow cytometry fluorophores are least susceptible to AF interference and that True Black can be used to remove AF signals from fixed tissue. Further, they point out the implications of these AF granules for complicating interpretation of microglial engulfment in their discussion. Hence, the novelty of the present study is reduced somewhat.

Response: We understand the reviewer’s point. However, none of the studies provided data that lipo-AF can confound engulfment analysis. As the reviewer mentioned, one study briefly mentioned it as a discussion point, but this was a relatively small point with no supporting data. Our manuscript expands upon this to show the extent to which lipo-AF can interfere with these analyses. It is so critical that the field performs these assays with rigor. In regards to the lack of novelty for the photobleaching protocol, we agree this has been demonstrated previously and we have removed “new” when referring to our protocol to avoid overselling. This is certainly not our intention. To increase the novelty, we developed our own device, which can be made cheaply and easily, to photobleach samples with similar efficacy to the commercial-grade photobleacher (**new Fig. 6 and new Supplementary Fig. 4**).

2) The direct examination of how photobleaching AF affects quantification of microglial synapse engulfment in cortex is novel. But this is explored in a relatively superficial manner. Does the concentration of VGlut2 antibody (and secondary antibody) used affect the degree to which AF granules are misinterpreted as microglial engulfment of synapses? Presumably the dimmer the antibody-labeled signal, the more one will need to turn up laser intensity / detector gain and the more AF signals will create problems. Similarly, performance of individual antibodies varies widely – is the degree to which AF introduces spurious engulfment results similar for synaptic terminals labeled with antibodies against VGlut1, synaptophysin, bassoon or similar? Are spurious engulfment results avoided even without photobleaching if 405 secondaries are used rather than red? How much are spurious engulfment results affected by postfixation length or how long brain sections have been in storage (and whether they were protected from light during storage and staining)? How much is the degree of spurious engulfment results affected by the imaging system? It would be of interest to know whether microglial AF is similarly problematic if tissue sections from the same mice, stained in the same way (+/- photobleaching) are imaged on systems with distinct optics, laser type and age, and detector type. Essentially – it feels like these authors would be in a unique position, given their expertise, to provide the field with more information about how prevalent spurious engulfment results may be and what are the parameters that most increase the risk of engulfment analyses being contaminated by AF granules. This is the type of information that would truly elevate this study above what has already been done in the field.

Response: We agree that there are numerous parameters that one could change to assess how lipo-AF can interfere with engulfment analysis. As the reviewer points out, every system, every antibody, every source of tissue will likely have varying degrees of lipo-AF signal. To further address this reviewer’s concern, we have performed the following:

1. We now show lipo-AF can be detected within microglia on the same microscope with 3 different laser powers (**new Fig. 1d-f**).

2. We now show on 3 different microscope set ups that lipo-AF can be detected (Fig. 1 and **new Supplementary Fig. 1**) and we write detailed descriptions of these microscope set ups in the revised methods.
3. We now show with two other synaptic markers and their degree of overlap with lipo-AF (**new Fig. 4d-i**).
4. The 405 channel could be used instead of photobleaching, but many antibodies do not provide a strong signal with these fluorophores and are not conducive to high quality staining. Therefore, the photobleaching method is a better option.
5. We now note in the updated text for the reviewer that our tissues all have different fixation times (mouse tissue is relatively short (hours) while human and marmoset are much longer (days)). This ranges from a relatively light fixation and short storage time for mouse tissue to a longer fixation and longer storage time for marmoset and human tissues. Also, mouse tissue was prepared as cryosections while human and marmoset were all FFPE tissue. Lipo-AF is detected in all tissues. We now note this in the revised text.

We tried our very best to test multiple different parameters that the reviewer noted. In all instances, we still detected lipo-AF.

3) If the main point of the paper is that AF material could be mistaken for engulfed material, it seems like they need to show this. Currently, they only show that photobleaching alters the amount of VGlut2 signal detected as being inside microglia at P90. Can they put the VGlut2 antibody in far red and leave the red channel open to show how many instances of “engulfed” VGlut2 puncta have an AF signal in the red channel? Of course, VGlut2 synaptic terminals could have been engulfed and trafficked to lysosomes that also have AF material.....but, together with the photobleaching experiments, this would directly demonstrate how AF signals could be mistaken for engulfment signals.

Response: This is a good point raised by the reviewer. We now provide a more comprehensive assessment of colocalization and quantify the amount of colocalization of VGlut2 with lipo-AF (**new Fig. 4a-c**).

4) TrueBlack can also be applied during blocking and prior to antibody staining. This would likely avoid the loss in P2RY12 signal that the authors observed. If this were true, it would greatly strengthen the manuscript to be able to compare removal of AF via photobleaching and removal of AF via TrueBlack and show whether these approaches are equally effective in rendering microglial VGlut2 engulfment analysis more robust (at least in rodent tissue). Did the authors try TrueBlack application prior to antibody staining?

Response: We appreciate the reviewer’s point. The issue with adding TrueBlack prior to immunostaining is that it requires detergent-free buffers, which is not always compatible with immunostaining. Instead, we provide a protocol for labs to make their own simple and inexpensive photobleacher (**new Fig. 6 and new Supplementary Fig. 4**).

5) The observation that microglia near to A-beta plaques potentially have less AF is surprising and also a novel observation. However, this, too, is explored only in a very cursory way. What % of plaques have microglia with little or no AF near them? How is this effect impacted by the size of the plaque? Is this

effect only noticed in microglia that are immediately adjacent to / potentially contacting plaques? What is the status of the lysosomes in the plaque-adjacent microglia with little or no AF?

Response: These are great points. We have since gone back and re-analyzed these data in a different way per the reviewer's comments. Because previous data were analyzed by manually thresholding individual cells and microglia around larger plaques are more difficult to segment, we had incorrectly biased our analysis towards IBA1+ cells surrounding smaller plaques where cell boundaries are more easily discernable. Instead, we now assess total volume within an area surrounding small and large plaques. Interestingly, microglia around smaller plaques have less lipo-AF burden and microglia around larger plaques have larger lipo-AF burden (**new Fig. 3c**). However, when you take the total average area of lipo-AF within all microglia around all plaques, there is no longer a significantly different lipo-AF burden in plaque-associated microglia from non-plaque associated microglia (**new Fig. 3b**).

Minor Concerns

1) Throughout the manuscript, the images appear to have had quite a bit of contrast introduced. The methods mention that images were "pre-processed" before analysis but what this involves isn't described. Some of the images also appear quite pixelated.

Response: We have now added more explanation and improved any image that appears pixelated. The latter may have been a consequence of the pdf assembly and compression.

2) Fig 1F, 2C, 3C, 3J, graphs seem redundant with the individual graphs that accompany them and that have the statistics and this causes some confusion (the reader is trying to determine if something different is being shown or just the same thing in different formats). This reviewer would also recommend giving these graphs different titles "negligible AF burden (0-0.1% cell vol)" "small AF burden (0.1-1% cell vol)" "moderate AF burden (1-2% cell vol)" "severe AF burden (>2.0% cell vol)" or similar. "% AF within microglia" at first seems like it's referring to how much AF is in the microglia versus how much is outside of the microglia.

Response: We agree. The reviewer makes an excellent suggestion to rename the titles of the graphs and we have done this in the updated manuscript. We have also moved the individual graphs to the supplement to avoid confusion.

3) In figure 4F, the authors should show the raw data – not just the ratio of detected engulfment with or without photobleaching. This would give greater transparency about the results and structure of the data.

Response: We have now added the raw data to **new Fig. 5k-n**.

4) It would be helpful to show some lower magnification images of VGlut2 staining (and possibly staining for some other punctate markers – e.g. Vglut1, synaptophysin, aggrecan) in P5 and P90 with and without photobleaching. This would give a better sense of how much the overall landscape of labeling with these markers is altered by photobleaching and removal of AF.

Response: We have now added more synaptic markers (**new Fig. 4d-i**). We have also added lower magnification images of VGlut2 with and without photobleaching (**new Supplementary Fig 5l-m**).

5) The manuscript seems to be overselling the finding that AF shows up first in microglia before other cells. To claim this more strongly, they would ideally look at the timecourse of AF increasing in microglia

and other cells in all the wavelengths, not just red, and with other ages between P90 and 24mo.

Response: We now include a timecourse of lipo-AF within microglia across multiple laser lines (**new Supplementary Fig. 3**). In all cases, microglia appear to accumulate lipo-AF first.

6) In the discussion it might be nice to focus less on the implications of lipofuscin for microglial biology (since this isn't a primary focus of the study). Instead, they could include discussion of strategies for accurate detection of AF (such as spectral unmixing). Is this a strategy that researchers could use in their systems to determine how much AF might be a concern for engulfment analyses? They could also discuss complementary approaches for detecting microglial engulfment of materials, such as EM analysis or immunohistochemical (non-fluorescent) imaging based methods. It would also be lovely to see a discussion of tools and approaches not yet available in the field to study microglial engulfment more robustly. TurboID targeted to microglial lysosomes to allow tagging and subsequent protein level analysis of material that is trafficked to microglial lysosomes?

Response: We believe that the next interesting direction of this manuscript is to study lipofuscin biology in microglia more directly. We also believe that some of the work showing lipo-AF accumulation in different brain areas and over aging adds interesting new biology. Therefore, we chose to keep this discussion in the manuscript. Still, this is an excellent suggestion to add a discussion of additional methods to study microglial synapse engulfment. Per the reviewer's suggestion, we now add a discussion of these additional methods.

Reviewer #3 (Remarks to the Author):

This work addresses an important gap in knowledge in microglial biology. The authors did an excellent job identifying a novel method to decrease and deplete immunostained tissue of auto fluorescent lipo-like AF's within microglia. This technique is even more valuable given their data suggesting lipo-AF's are confounding other typical engulfment contents of microglia, especially during crucial periods of synaptic pruning earlier in development. Overall, this work and technique is important to better understand the substrates microglia are clearing without any auto fluorescent interference. Implementing this technique in labs that study phagocytosis. would be crucial as to not misinterpret the engulfment contents of microglia in general, across development, and in instances of neurodegenerative disease. I have only a few minor concerns and questions outlined below.

Response: We thank the reviewer for their positive feedback and for appreciating the importance of the manuscript. We have done our best to address their concerns raised below to improve the manuscript. Overall minor concerns:

General:

Please report/display individual data points overlaid onto bar graphs in all %AF graphs like those in Figure 1f-1k. This is important given interest in heterogeneity in microglia in many brain regions, beyond simple transparency in data reporting.

Response: We have now included individual data points.

- In the instances where lipo-AF was not co-localized within CD68+ lysosomes or Iba1+ microglia (Figure 1k), Is it possible that the older microglia are reaching a maximum phagocytic capacity?

Response: We have now looked into this more thoroughly. Microglia still accumulate more lipo-AF with age compared to younger timepoints. However, the increased lipo-AF outside of microglia at older ages is largely neuronal and some is in border macrophages (**new Supplementary Fig. 2f-i**). See also response to point below.

- Is there another subset of microglia that are not Iba1+, or maybe another glial or macrophage population that are able to engulf the lipo-AF in the older animals? Or are the microglia themselves potentially undergoing cell death or apoptosis and the accumulations of lipo-AF that are left behind following this? The amount of unengulfed excess lipo-AF at the 24m age is very interesting, and worth exploring whether another non-Iba1+ cell type is engulfing the lipo-AF outside of the microglia. Could you use other macrophage/microglia markers to label additional cells that could be engulfing these?

Response: These are all interesting points. We emphasize that microglia still have a very large and significantly increased lipo-AF burden at older ages. However, other cells start to accumulate lipo-AF with age. As microglia are smaller in number compared to other cell types, lipo-AF burden outside microglia is higher at older ages. We now show that this lipo-AF outside microglia is largely in neurons and a small amount is in border macrophages (**new Supplementary Fig. 2f-i**). As neurons are not thought to be phagocytic, we suspect this could be through, for example, failed autophagy and/or changes of proteostasis and lipolysis over aging. We now include this in the discussion in the revised manuscript.

- In Figure 1K, was there a significant difference between p5 and p15 & p30? It is interesting that the

lipo-AF volume appears higher in earlier timepoints and then tapers, similar to crucial timepoints of synaptic pruning in the developing CNS. Maybe this could also be younger microglia reaching phagocytic capacity with synaptic debris and they cannot uptake as much of the lipo-AF?

Response: There is no significant difference in lipo-AF burden within microglia between P5, P15, and P30. However, we do suspect that the minimal amount of lipo-AF within microglia may result from engulfed cellular material, including synapses and myelin. Indeed, we now show that the lipo-AF burden is most significant in white matter at younger ages (**new Fig. 2d,e**). We also include a more in-depth discussion of these results in the updated text.

Figure 2:

No comments

Figure 3:

No comments, the comparison of the two distinct techniques and their strengths and weaknesses is thoroughly addressed in the figure and discussion sections.

Figure 4:

- This is a super exciting technique and finding that lipo-AF can impact the interpretation of microglial contents. Is this something that could be (or should be) used for tissue earlier in development? Or does the photobleaching have an age threshold in which it works best between P5 and P90?

Response: We thank the reviewers for their enthusiasm for the technique. We suspect photobleaching can be used at any age and suggest that this should be determined empirically. We emphasize though that there is really minimal to no lipo-AF at early postnatal ages.

Figure 5:

No comments, very important cross-species comparison.

REVIEWERS' COMMENTS

Reviewer #1 (Remarks to the Author):

The authors made a big effort to address the concerns of this reviewer. The additional experiments and amendments significantly improved the quality and relevance of paper. I find the regional lipo-AF accumulation of interest. I have no further concerns.

Reviewer #2 (Remarks to the Author):

In this revised manuscript, the authors have added experiments, analyses, and text that deepen and expand the original findings. The study is greatly strengthened and my concerns have been largely addressed. A few minor concerns are listed below. We still have reservation about whether the overall findings are sufficiently novel/impactful to merit publication in the present journal. However, we also agree with the idea that this "cautionary tale" should ideally be published in a forum with wide readership to help make studies of microglial phagocytic function more rigorous throughout the field.

minor concerns

1) In figure 1D, which shows lipo-AF with different laser intensities, it would be ideal for the images to come from the same field of view. When they come from different fields of view (and possibly different animals?), other factors such as perfusion quality, time of tissue storage, and inter-animal variation in the biological factors that drive formation of these aggregates will also presumably affect the lipo-AF intensity.

2) Figure 2C, F, H - suggest relabeling y axis of graphs to "% FOV lipo-AF inside microglia" or similar. The current label suggests that there is still a focus on microglial AF and what percentage of that AF has certain characteristics (such as being inside CD68 lysosomes). There is also some confusion / inconsistency in the figure legend. It says that 2C shows "lipo-AF per field of view inside microglia" and F and H say "lipo-AF per field of view OUTSIDE microglia." Is this a typo for F and H? A similar issue with potentially confusing axis labels is present in Sup. Fig. 3B and C.

3) For the quantification in figure 3C, doesn't there need to be a normalization for "microglial area?" The larger the plaque is, the larger the number of microglia that will be surrounding that plaque, and then, of course, the total summed volume of lipo-AF in plaque associated microglia will be larger. It doesn't really answer the question of whether microglia near to small plaques have more lipo-AF relative to their cell volume compared to microglia near large plaques. It isn't necessary to segment individual microglia, but still seems necessary to quantify the total "microglial area" that is surrounding a particular plaque and then use this to normalize the total volume of lipo-AF that are in those plaque associated microglia.

4) Figure 4F, it seems like the image with Iba1 has been placed on top, but according to the labels, the

image that has Iba1 should be below (as it is in 4D and 4H).

5) It is slightly confusing to interpret what is being shown in 4E,G and I. Does the Vglut2/1/homer IN signal represent a synaptic signal inside microglia that is clearly NOT lipo-AF? And is the lipo-AF intensity representing the intensity of the autofluorescence in 635nm? Or does that refer to the intensity of the synaptic signal (Vglut2/1/homer) when it happens to be colocalized with lipo-AF? It seems that the most important info here is the difference in signal intensity in the red channel between synaptic elements inside microglia that represent true synaptic elements (no colocalization with lipo-AF in 635) versus spurious synaptic elements (signal in red that is colocalized with lipo-AF in 635).

Reviewer #3 (Remarks to the Author):

All of my concerns have been addressed.

We thank the reviewers for their careful and thorough review of the manuscript, which has only improved as a result. Any changes are highlighted in red text in the revised manuscript.

Reviewer #1 (Remarks to the Author):

In this manuscript entitled ‘Lipofuscin-like autofluorescence within microglia and its impact on studying microglial engulfment’, Stillman and colleagues aimed to develop a method to quench autofluorescence (AF) signal in microglia that could potentially confound downstream analyses. The authors first characterized AF in the somatosensory cortex at different timepoints and found that microglia exhibit more AF over time. Further, the authors hypothesize lipofuscin within lysosomes to be the main source of AF, that could potentially interfere with popular histological analyses within microglia such as the synaptic engulfment assay, introduced by the senior author during her postdoctoral training at the Stevens lab (Schafer et al., 2014). Indeed, data demonstrated in the current manuscript suggests that lipofuscin- AF could potentially be misinterpreted as false-positive synaptic immunoreactivity when analysed using the yellow/green (561nm) and far-red laser, but not UV laser. Of interest, the authors did not find significant increase of AF in microglia in the 5xFAD model. The authors finally adapted their model, based on prior quenching of any background using either chemical or light-based bleaching, to mouse, marmoset and human brain tissue.

The manuscript could be of potential interest to the microglia community because it offers a protocol that could improve reproducibility of immunofluorescence-based experiments. However, experimental shortcomings and the general lack of originality limits my enthusiasm for this manuscript. The authors introduce a model for an artefact that one can potentially overcome by using the appropriate controls and careful considerations, for example only secondary antibody incubation control, adjustment for laser power etc. In fact, many research laboratories do not have access to Merscope equipment suggested by the authors. Evidence provided by the authors that AF could interfere with downstream analyses (engulfment assay, but are there any others?) and that signal quenching could improve data interpretation, should be more convincing and robust. Please find detailed points and suggestions below:

Response: We thank the reviewer for their careful review of our manuscript and for appreciating the value to the community. We have done our best to address the reviewer’s experimental critique below and our manuscript has improved as a result. In regards to originality, it is true that microglial lipofuscin accumulation is known and that there are existing protocols for quenching autofluorescence. We are of the opinion that the importance of this manuscript is more in its “cautionary tale” to the field that this lipo-AF can interfere with synapse engulfment, as well as engulfment of other substrates. These sorts of studies are so critical to move the field forward in an arena where it will be read broadly. We also think the photobleaching protocol is extremely useful to labs working inside and outside of the brain and we appreciate the reviewer’s concern about the MERSCOPE photobleacher’s accessibility. We, therefore, now provide a protocol to build your own inexpensive, simple photobleaching device (**New Fig. 6 and new Supplementary Fig. 4**).

Major points/suggestions:

1) In Fig. 4, the authors demonstrate that synapse immunoreactivity within microglia is comparable to

the percentage of AF within microglia of an adjacent brain slice, suggesting that synapse immunoreactivity could potentially be misinterpreted due to AF. This is not very convincing, and the authors should provide more rationale to motivate different labs to adapt the suggested protocol. For example, could the authors design an experiment where colocalization is measured between synapses and AF, by immunostaining for synapses using fluorescent dyes in the red spectra (for example, alexa 647) and AF excited at 488nm? There does not seem to be a significant increase in autofluorescence using the red (638 nm) laser compared to the UV laser (Fig 1b and c). I think the obtained data could be more informative and a better measure for potential false-positive synapse staining caused by AF. In summary, if the authors could demonstrate that AF could lead to false-positive synapse immunoreactivity in a more convincing way, this would greatly contribute to the significance of the manuscript.

Response: We now provide a more comprehensive assessment of colocalization of lipo-AF with VGLuT2 (**new Fig. 4a-c**). We also show in **Fig. 5k-n** that apparent engulfed VGLuT2 in adult animals is eliminated after photobleaching. This last experiment confirms that lipo-AF can be confused for anti-synaptic staining inside (but not outside) microglia. We further show analyses with other synaptic markers (**new Fig. 4d-i**)

2) The authors demonstrate that the quenching protocol does not compromise P2Y12 immunostaining. Could the authors perform additional experiments to demonstrate the preservation of fluorescent signal of microglial markers of interest within the marmoset and human tissue? It would also be important to demonstrate that this protocol does not compromise lysosomal CD68+ and LAMP1+ immunostaining when considering photobleaching prior to synaptic engulfment assay. In this regard, CD68 staining is demonstrated in figure 4D and 4E after treatment, so I assume signal will not be quenched?

Response: To address this concern, we have shown that anti-CD68 signal is not significantly reduced with our homemade photobleacher (**new Supplementary Fig. 5C**), anti-IBA1 does not quench with mouse, human MS and marmoset tissue (**new Supplementary Fig. 6d-k**), and anti-GFP signal does not significantly quench (**new Supplementary Fig. 6a-b**).

3) It would be of great interest to characterise whether there is region-specific appearance of AF+ microglia. For example, the authors mention a subset of AF+ microglia (line 88) that appears during postnatal development. In contrast, in the 5xFAD model, there seems to be no spatial heterogeneity in AF+ microglia located either close or further away from plaques. Could the authors provide more experimental information on where exactly AF+ microglia are located? For example, is there difference between white and grey matter regions? Further, it would be informative to provide an overview of AF+ microglia abundance in regions that are relevant to microglia-mediated synaptic engulfment (for example, barrel cortex, retinogeniculate system,...). Finally, it would be of interest for the manuscript if the authors can speculate more on AF+ microglia heterogeneity in the discussion part. For example, do the authors speculate that certain microglia subpopulations are more prone to AF accumulation? In line, but probably not the scope of this manuscript, it would be interesting to probe whether the AF+ microglia acquire a distinct functional phenotype due to potential lipofuscin accumulation.

Response: This is an excellent point. We now show that microglia AF accumulates first in white matter vs. gray matter (**new Fig. 2 and Supplementary Fig. 2**). However, in aged mice, AF accumulates throughout microglia in white and gray matter regions to similar levels. We further expand the discussion to speculate on why white matter microglia may accumulate lipo-AF sooner than gray matter.

Minor points:

-I assume background signal and AF is in part determined by laser power, so it would be useful if the authors could mention the laser power settings in the materials and method part.

Response: We now include a more detailed description of our microscope settings. We further add new data that lipo-AF can be detected at different laser powers with the same microscope (**new Fig. 1d-f**) and it can be detected with other microscopes (**new Supplementary Fig 1**).

-Please provide dot plots or show individual datapoints in each figure for transparency. Please clearly indicate what one data point represents in each figure legend (1 datapoint= one microglia, one region of interest, 1 animal...).

Response: These individual data points have now been added and we define the n's as animals (1 data point=1 animal).

-With regards to the supplementary figure, could the authors provide representative lysosomal stainings for each time point assessed in figure 1? It will be of interest to see whether the ratio AF volume to lysosomes/ other cellular compartments varies over the different time points.

Response: We now added representative CD68+ lysosomal stainings (**new Fig. 2a,d**) and quantify this staining for each time point (**new Supplementary Fig. 2c,e**).

-Related to my previous point, it seems that microglia are not the main source of the AF at later time points, an interesting observation mentioned but not further investigated by the authors. For example, by looking at figure 1e-24mo, it seems to be that microglia represent only a small proportion of the AF+ cells. Could the authors further speculate on this in the discussion?

Response: We agree. This is interesting. We emphasize that microglia still have a very large and significantly increased lipo-AF burden at older ages. However, other cells do start to accumulate lipo-AF with age. As microglia are smaller in number compared to other cell types, total lipo-AF outside microglia is higher at older ages. We now show that most of lipo-AF outside of microglia at older ages is from neurons (**new Supplementary Fig. 2h-i**). We add further discussion of this timeline of lipo-AF accumulation in microglia vs. other cells in the revised manuscript discussion.

-I assume the protocol quenches any endogenous fluorescence (for example GFP or TdTomato, often used in microglia-based reporter mice), but would it be possible to retain the signal by staining for anti-GFP or anti-RFP, for example?

Response: Yes, with antibody staining amplification, endogenous fluorescence can still be visualized after photobleaching. We added these data to the revised manuscript (**new Supplementary Fig. 5a-b**).

-It is not clear from the main text whether the synaptic engulfment assay represents VGLUT2 staining within IBA1+ cells or CD68+ IBA1+ lysosomes. Could the authors make this more clear in the text?

Response: This is engulfment assay represent VGluT2 within CD68+/IBA1+ lysosomes. We now clarify this in the text.

-Line 72-73, please provide references for the following sentence "We focused our imaging in the somatosensory cortex and neighboring visual and auditory cortices as this was a large region that could be easily identified across ages and is known to undergo neurodegeneration REF"

Response: Thank you for catching this. This sentence has now been omitted in the revised manuscript.

-Line 85: Liop-AF should be Lipo-AF

Response: This has been changed. Thank you for pointing this out.

-Can the authors touch on AF within border-associated macrophages? These cells are receiving increased attention recently, so it would be of interest to demonstrate whether BAM are a potential source of lipofuscin-AF or not.

Response: We have now added new data showing and quantifying AF within border-associated macrophages (**new Supplementary Fig. 2f-g**).

Reviewer #2 (Remarks to the Author):

In this manuscript, the authors examine a key aspect of microglial biology – phagocytic engulfment of cellular material – and highlight how light microscopic study of this microglial function can be confounded by autofluorescent (AF) material within these cells. As the authors themselves note, a growing number of studies point to microglial engulfment of synaptic material, extracellular matrix, myelin, dying cells, and pathological protein aggregates as playing a key role in shaping circuit function during development, health and disease. Many of these data rely on light microscopy and colocalization of potentially engulfed material with microglial lysosomes. Yet, the AF material in microglia is also localized to lysosomes, indicating that this approach cannot unequivocally confirm microglial engulfment.

Here, the authors analyze at what age AF signals appear in cortical microglia and show that TrueBlack or photobleaching can be used to remove these AF signals. They examine whether microglia near to Aβ plaques have exacerbated AF accumulation, and find, surprisingly, that they do not. Most importantly, they examine how incorporation of photobleaching approaches impacts quantification of microglial synapse engulfment in early postnatal and young adult mice. Their results indicate that much of the putative engulfment observed in young adult mice is likely a “false-positive” result due to AF accumulation.

On the whole the manuscript is well-written, the figures clearly presented, and the topic of accurate analysis of microglial engulfment is an important one for the field. However, the analyses presented are somewhat superficial and there are key concerns about the novelty as other studies have already detailed how to use photobleaching and other approaches to remove AF signals from brain tissue.

Response: We greatly appreciate the reviewer’s feedback and for appreciating the value of our study to the community. We have done our best to address the reviewer’s concerns below. In regards to novelty, it is true that microglial lipofuscin accumulation is known and that there are existing protocols for quenching autofluorescence. We are of the opinion that, the most important aspect of this manuscript, as the reviewer also points out, is more in its “cautionary tale” to the field. That is, this lipo-AF can interfere with synapse engulfment, as well as engulfment of other substrates. These sorts of studies are so critical to move the field forward and it is critical that they are published in an arena where it will be read broadly. Also, while photobleaching has been shown to rid of autofluorescence in brain tissues before, this has never been shown across species. We now include in the revised proposal, imaging of lipofuscin with different laser intensities (**new Fig. 1d-f**), different microscope set-ups (**new Supplementary Fig. 1**), and different synaptic antibodies (**new Fig. 4d-i**). Also, in the revised manuscript, we provide a protocol (**new Fig. 6 and Supplementary Fig. 4**) to build your own inexpensive photobleaching device with similar photobleaching efficacy as the MERFISH photobleacher.

Major Concerns

1) This reviewer’s main concern is linked to several other recent publications (most of which the authors cite) that have provided detailed methods for detecting and removing AF from light microscopic analysis of rodent, NHP, and human brain tissue PMID: 31790435, PMID: 31379520, PMID: 28892031, PMID: 23994358, PMID: 35361108. This includes both chemical methods like True Black and photobleaching

approaches. Hence, it feels like an overstatement to say that the current manuscript provides the field with a “new protocol” (line 189) that can be used to get rid of AF. Moreover, use of these approaches has already been demonstrated in human tissue PMID: 28892031. One of the above publications (PMID: 35361108) explores AF specifically in microglia and analyzes how this impacts flow cytometry and light microscopy in fixed tissue, acute brain sections, and in vivo imaging. They show which flow cytometry fluorophores are least susceptible to AF interference and that True Black can be used to remove AF signals from fixed tissue. Further, they point out the implications of these AF granules for complicating interpretation of microglial engulfment in their discussion. Hence, the novelty of the present study is reduced somewhat.

Response: We understand the reviewer’s point. However, none of the studies provided data that lipo-AF can confound engulfment analysis. As the reviewer mentioned, one study briefly mentioned it as a discussion point, but this was a relatively small point with no supporting data. Our manuscript expands upon this to show the extent to which lipo-AF can interfere with these analyses. It is so critical that the field performs these assays with rigor. In regards to the lack of novelty for the photobleaching protocol, we agree this has been demonstrated previously and we have removed “new” when referring to our protocol to avoid overselling. This is certainly not our intention. To increase the novelty, we developed our own device, which can be made cheaply and easily, to photobleach samples with similar efficacy to the commercial-grade photobleacher (**new Fig. 6 and new Supplementary Fig. 4**).

2) The direct examination of how photobleaching AF affects quantification of microglial synapse engulfment in cortex is novel. But this is explored in a relatively superficial manner. Does the concentration of VGlut2 antibody (and secondary antibody) used affect the degree to which AF granules are misinterpreted as microglial engulfment of synapses? Presumably the dimmer the antibody-labeled signal, the more one will need to turn up laser intensity / detector gain and the more AF signals will create problems. Similarly, performance of individual antibodies varies widely – is the degree to which AF introduces spurious engulfment results similar for synaptic terminals labeled with antibodies against VGlut1, synaptophysin, bassoon or similar? Are spurious engulfment results avoided even without photobleaching if 405 secondaries are used rather than red? How much are spurious engulfment results affected by postfixation length or how long brain sections have been in storage (and whether they were protected from light during storage and staining)? How much is the degree of spurious engulfment results affected by the imaging system? It would be of interest to know whether microglial AF is similarly problematic if tissue sections from the same mice, stained in the same way (+/- photobleaching) are imaged on systems with distinct optics, laser type and age, and detector type. Essentially – it feels like these authors would be in a unique position, given their expertise, to provide the field with more information about how prevalent spurious engulfment results may be and what are the parameters that most increase the risk of engulfment analyses being contaminated by AF granules. This is the type of information that would truly elevate this study above what has already been done in the field.

Response: We agree that there are numerous parameters that one could change to assess how lipo-AF can interfere with engulfment analysis. As the reviewer points out, every system, every antibody, every source of tissue will likely have varying degrees of lipo-AF signal. To further address this reviewer’s concern, we have performed the following:

1. We now show lipo-AF can be detected within microglia on the same microscope with 3 different laser powers (**new Fig. 1d-f**).

2. We now show on 3 different microscope set ups that lipo-AF can be detected (Fig. 1 and **new Supplementary Fig. 1**) and we write detailed descriptions of these microscope set ups in the revised methods.
3. We now show with two other synaptic markers and their degree of overlap with lipo-AF (**new Fig. 4d-i**).
4. The 405 channel could be used instead of photobleaching, but many antibodies do not provide a strong signal with these fluorophores and are not conducive to high quality staining. Therefore, the photobleaching method is a better option.
5. We now note in the updated text for the reviewer that our tissues all have different fixation times (mouse tissue is relatively short (hours) while human and marmoset are much longer (days)). This ranges from a relatively light fixation and short storage time for mouse tissue to a longer fixation and longer storage time for marmoset and human tissues. Also, mouse tissue was prepared as cryosections while human and marmoset were all FFPE tissue. Lipo-AF is detected in all tissues. We now note this in the revised text.

We tried our very best to test multiple different parameters that the reviewer noted. In all instances, we still detected lipo-AF.

3) If the main point of the paper is that AF material could be mistaken for engulfed material, it seems like they need to show this. Currently, they only show that photobleaching alters the amount of VGlut2 signal detected as being inside microglia at P90. Can they put the VGlut2 antibody in far red and leave the red channel open to show how many instances of “engulfed” VGlut2 puncta have an AF signal in the red channel? Of course, VGlut2 synaptic terminals could have been engulfed and trafficked to lysosomes that also have AF material.....but, together with the photobleaching experiments, this would directly demonstrate how AF signals could be mistaken for engulfment signals.

Response: This is a good point raised by the reviewer. We now provide a more comprehensive assessment of colocalization and quantify the amount of colocalization of VGlut2 with lipo-AF (**new Fig. 4a-c**).

4) TrueBlack can also be applied during blocking and prior to antibody staining. This would likely avoid the loss in P2RY12 signal that the authors observed. If this were true, it would greatly strengthen the manuscript to be able to compare removal of AF via photobleaching and removal of AF via TrueBlack and show whether these approaches are equally effective in rendering microglial VGlut2 engulfment analysis more robust (at least in rodent tissue). Did the authors try TrueBlack application prior to antibody staining?

Response: We appreciate the reviewer’s point. The issue with adding TrueBlack prior to immunostaining is that it requires detergent-free buffers, which is not always compatible with immunostaining. Instead, we provide a protocol for labs to make their own simple and inexpensive photobleacher (**new Fig. 6 and new Supplementary Fig. 4**).

5) The observation that microglia near to A-beta plaques potentially have less AF is surprising and also a novel observation. However, this, too, is explored only in a very cursory way. What % of plaques have microglia with little or no AF near them? How is this effect impacted by the size of the plaque? Is this

effect only noticed in microglia that are immediately adjacent to / potentially contacting plaques? What is the status of the lysosomes in the plaque-adjacent microglia with little or no AF?

Response: These are great points. We have since gone back and re-analyzed these data in a different way per the reviewer's comments. Because previous data were analyzed by manually thresholding individual cells and microglia around larger plaques are more difficult to segment, we had incorrectly biased our analysis towards IBA1+ cells surrounding smaller plaques where cell boundaries are more easily discernable. Instead, we now assess total volume within an area surrounding small and large plaques. Interestingly, microglia around smaller plaques have less lipo-AF burden and microglia around larger plaques have larger lipo-AF burden (**new Fig. 3c**). However, when you take the total average area of lipo-AF within all microglia around all plaques, there is no longer a significantly different lipo-AF burden in plaque-associated microglia from non-plaque associated microglia (**new Fig. 3b**).

Minor Concerns

1) Throughout the manuscript, the images appear to have had quite a bit of contrast introduced. The methods mention that images were "pre-processed" before analysis but what this involves isn't described. Some of the images also appear quite pixelated.

Response: We have now added more explanation and improved any image that appears pixelated. The latter may have been a consequence of the pdf assembly and compression.

2) Fig 1F, 2C, 3C, 3J, graphs seem redundant with the individual graphs that accompany them and that have the statistics and this causes some confusion (the reader is trying to determine if something different is being shown or just the same thing in different formats). This reviewer would also recommend giving these graphs different titles "negligible AF burden (0-0.1% cell vol)" "small AF burden (0.1-1% cell vol)" "moderate AF burden (1-2% cell vol)" "severe AF burden (>2.0% cell vol)" or similar. "% AF within microglia" at first seems like it's referring to how much AF is in the microglia versus how much is outside of the microglia.

Response: We agree. The reviewer makes an excellent suggestion to rename the titles of the graphs and we have done this in the updated manuscript. We have also moved the individual graphs to the supplement to avoid confusion.

3) In figure 4F, the authors should show the raw data – not just the ratio of detected engulfment with or without photobleaching. This would give greater transparency about the results and structure of the data.

Response: We have now added the raw data to **new Fig. 5k-n**.

4) It would be helpful to show some lower magnification images of VGlut2 staining (and possibly staining for some other punctate markers – e.g. Vglut1, synaptophysin, aggrecan) in P5 and P90 with and without photobleaching. This would give a better sense of how much the overall landscape of labeling with these markers is altered by photobleaching and removal of AF.

Response: We have now added more synaptic markers (**new Fig. 4d-i**). We have also added lower magnification images of VGlut2 with and without photobleaching (**new Supplementary Fig 5l-m**).

5) The manuscript seems to be overselling the finding that AF shows up first in microglia before other cells. To claim this more strongly, they would ideally look at the timecourse of AF increasing in microglia

and other cells in all the wavelengths, not just red, and with other ages between P90 and 24mo.

Response: We now include a timecourse of lipo-AF within microglia across multiple laser lines (**new Supplementary Fig. 3**). In all cases, microglia appear to accumulate lipo-AF first.

6) In the discussion it might be nice to focus less on the implications of lipofuscin for microglial biology (since this isn't a primary focus of the study). Instead, they could include discussion of strategies for accurate detection of AF (such as spectral unmixing). Is this a strategy that researchers could use in their systems to determine how much AF might be a concern for engulfment analyses? They could also discuss complementary approaches for detecting microglial engulfment of materials, such as EM analysis or immunohistochemical (non-fluorescent) imaging based methods. It would also be lovely to see a discussion of tools and approaches not yet available in the field to study microglial engulfment more robustly. TurboID targeted to microglial lysosomes to allow tagging and subsequent protein level analysis of material that is trafficked to microglial lysosomes?

Response: We believe that the next interesting direction of this manuscript is to study lipofuscin biology in microglia more directly. We also believe that some of the work showing lipo-AF accumulation in different brain areas and over aging adds interesting new biology. Therefore, we chose to keep this discussion in the manuscript. Still, this is an excellent suggestion to add a discussion of additional methods to study microglial synapse engulfment. Per the reviewer's suggestion, we now add a discussion of these additional methods.

Reviewer #3 (Remarks to the Author):

This work addresses an important gap in knowledge in microglial biology. The authors did an excellent job identifying a novel method to decrease and deplete immunostained tissue of auto fluorescent lipo-like AF's within microglia. This technique is even more valuable given their data suggesting lipo-AF's are confounding other typical engulfment contents of microglia, especially during crucial periods of synaptic pruning earlier in development. Overall, this work and technique is important to better understand the substrates microglia are clearing without any auto fluorescent interference. Implementing this technique in labs that study phagocytosis. would be crucial as to not misinterpret the engulfment contents of microglia in general, across development, and in instances of neurodegenerative disease. I have only a few minor concerns and questions outlined below.

Response: We thank the reviewer for their positive feedback and for appreciating the importance of the manuscript. We have done our best to address their concerns raised below to improve the manuscript. Overall minor concerns:

General:

Please report/display individual data points overlaid onto bar graphs in all %AF graphs like those in Figure 1f-1k. This is important given interest in heterogeneity in microglia in many brain regions, beyond simple transparency in data reporting.

Response: We have now included individual data points.

- In the instances where lipo-AF was not co-localized within CD68+ lysosomes or Iba1+ microglia (Figure 1k), Is it possible that the older microglia are reaching a maximum phagocytic capacity?

Response: We have now looked into this more thoroughly. Microglia still accumulate more lipo-AF with age compared to younger timepoints. However, the increased lipo-AF outside of microglia at older ages is largely neuronal and some is in border macrophages (**new Supplementary Fig. 2f-i**). See also response to point below.

- Is there another subset of microglia that are not Iba1+, or maybe another glial or macrophage population that are able to engulf the lipo-AF in the older animals? Or are the microglia themselves potentially undergoing cell death or apoptosis and the accumulations of lipo-AF that are left behind following this? The amount of unengulfed excess lipo-AF at the 24m age is very interesting, and worth exploring whether another non-Iba1+ cell type is engulfing the lipo-AF outside of the microglia. Could you use other macrophage/microglia markers to label additional cells that could be engulfing these?

Response: These are all interesting points. We emphasize that microglia still have a very large and significantly increased lipo-AF burden at older ages. However, other cells start to accumulate lipo-AF with age. As microglia are smaller in number compared to other cell types, lipo-AF burden outside microglia is higher at older ages. We now show that this lipo-AF outside microglia is largely in neurons and a small amount is in border macrophages (**new Supplementary Fig. 2f-i**). As neurons are not thought to be phagocytic, we suspect this could be through, for example, failed autophagy and/or changes of proteostasis and lipolysis over aging. We now include this in the discussion in the revised manuscript.

- In Figure 1K, was there a significant difference between p5 and p15 & p30? It is interesting that the

lipo-AF volume appears higher in earlier timepoints and then tapers, similar to crucial timepoints of synaptic pruning in the developing CNS. Maybe this could also be younger microglia reaching phagocytic capacity with synaptic debris and they cannot uptake as much of the lipo-AF?

Response: There is no significant difference in lipo-AF burden within microglia between P5, P15, and P30. However, we do suspect that the minimal amount of lipo-AF within microglia may result from engulfed cellular material, including synapses and myelin. Indeed, we now show that the lipo-AF burden is most significant in white matter at younger ages (**new Fig. 2d,e**). We also include a more in-depth discussion of these results in the updated text.

Figure 2:

No comments

Figure 3:

No comments, the comparison of the two distinct techniques and their strengths and weaknesses is thoroughly addressed in the figure and discussion sections.

Figure 4:

- This is a super exciting technique and finding that lipo-AF can impact the interpretation of microglial contents. Is this something that could be (or should be) used for tissue earlier in development? Or does the photobleaching have an age threshold in which it works best between P5 and P90?

Response: We thank the reviewers for their enthusiasm for the technique. We suspect photobleaching can be used at any age and suggest that this should be determined empirically. We emphasize though that there is really minimal to no lipo-AF at early postnatal ages.

Figure 5:

No comments, very important cross-species comparison.